# Improving Robustness using Generated Data

**Sven Gowal\*, Sylvestre-Alvise Rebuffi\*, Olivia Wiles,**
**Florian Stimberg**, **Dan Calian** and **Timothy Mann**
DeepMind, London
{sgowal,sylvestre}@deepmind.com

## Abstract

Recent work argues that robust training requires substantially larger datasets than those required for standard classification. On CIFAR-10 and CIFAR-100, this translates into a sizable robust-accuracy gap between models trained solely on data from the original training set and those trained with additional data extracted from the "80 Million Tiny Images" dataset (80M-TI). In this paper, we explore how generative models trained solely on the original training set can be leveraged to artificially increase the size of the original training set and improve adversarial robustness to $\ell_p$ norm-bounded perturbations. We identify the sufficient conditions under which incorporating additional generated data can improve robustness, and demonstrate that it is possible to significantly reduce the robust-accuracy gap to models trained with additional real data. Surprisingly, we show that even the addition of non-realistic random data (generated by Gaussian sampling) can improve robustness. We evaluate our approach on CIFAR-10, CIFAR-100, SVHN and TINYIMAGENET against $\ell_\infty$ and $\ell_2$ norm-bounded perturbations of size $\epsilon = 8/255$ and $\epsilon = 128/255$, respectively. We show large absolute improvements in robust accuracy compared to previous state-of-the-art methods. Against $\ell_\infty$ norm-bounded perturbations of size $\epsilon = 8/255$, our models achieve 66.10% and 33.49% robust accuracy on CIFAR-10 and CIFAR-100, respectively (improving upon the state-of-the-art by +8.96% and +3.29%). Against $\ell_2$ norm-bounded perturbations of size $\epsilon = 128/255$, our model achieves 78.31% on CIFAR-10 (+3.81%). These results beat most prior works that use external data.

## 1 Introduction

Neural networks are being deployed in a wide variety of applications ranging from ranking content on the web [15] to autonomous driving [5] via medical diagnostics [22]. It has become increasingly important to ensure that deployed models are robust and generalize to various input perturbations. Unfortunately, the addition of imperceptible adversarial perturbations can cause neural networks to make incorrect predictions [9, 10, 27, 44, 64]. There has been a lot of work on understanding and generating adversarial perturbations [1, 4, 10, 64], and on building defenses that are robust to such perturbations [27, 49, 60, 82]. We note that while robustness and invariance to input perturbations is crucial to the deployment of machine learning models in various applications, it can also have broader negative impacts to society such as hindering privacy [63] or increasing bias [68].

The adversarial training procedure proposed by Madry et al. [49] feeds adversarially perturbed examples back into the training data. It is widely regarded as one of the most successful method to train robust deep neural networks [30], and it has been augmented in different ways – with changes in the attack procedure [25], loss function [50, 82] or model architecture [76, 85]. We highlight the works by Carmon et al. [11], Najafi et al. [51], Uesato et al. [72], Zhai et al. [80] who simultaneously proposed the use of additional unlabeled external data. While the addition of external data helped boost robust accuracy by a large margin, progress in the setting without additional data has slowed

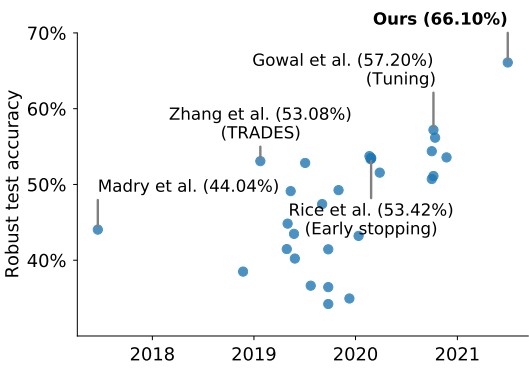

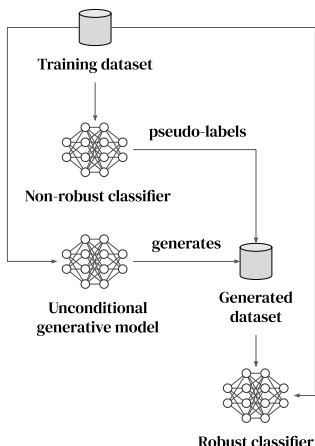

Figure 1: Robust accuracy of models against AU-TOATTACK [16] on CIFAR-10 with $\ell_\infty$ perturbations of size $8/255$ displayed in publication order. Our method explores how generated data can be used to improve robust accuracy by +8.96% without using any additional external data. This constitutes the largest jump in robust accuracy in this setting. Our best model reaches a robust accuracy of 66.10% against AA+MT [30].

Figure 2: Overview of our approach. Our method initially trains a generative model and a non-robust classifier. The non-robust classifier is used to provide pseudo-labels to the generated data. Finally, generated and original training data are combined to train a robust classifier.

(see Fig. 1). On CIFAR-10 [42] against $\ell_\infty$ perturbations of size $\epsilon = 8/255$, the best known model obtains a robust accuracy of 65.87% when using additional data. The same model obtains a robust accuracy of 57.14% without this data [30]. As a result, we ask ourselves whether it is possible to leverage the information contained in the original training set to a greater extent. This manuscript challenges the status-quo. To the contrary of standard training where it is widely believed that generative models lack diversity and that the samples they produce cannot be used to train better classifiers [59], we demonstrate both theoretically and experimentally that these generated samples can be used to improve robustness (using the approach described in Fig. 2 and Sec. 3.3). We make the following contributions:

- We demonstrate in Sec. 3.2 that it is possible to use low-quality random inputs (sampled from a conditional Gaussian fit of the training data) to improve robust accuracy on CIFAR-10 against $\ell_\infty$ perturbations of size $\epsilon = 8/255$ (+0.93% on a WRN-28-10) and provide a justification and sufficient conditions in Sec. 4.

- We leverage higher quality generated inputs (i.e., inputs generated by generative models solely trained on the original data), and study four recent generative models: the Denoising Diffusion Probabilistic Model (DDPM) [36], StyleGAN2 [40], BigGAN [7] and the Very Deep Variational Auto-Encoder (VDVAE) [14] (Sec. 5). We show that DDPM samples cover most closely the real data distribution (as measured by the distance to the test set in the Inception feature space).

- Using images generated by the DDPM allows us to reach a robust accuracy of 66.10% on CIFAR-10 against $\ell_\infty$ perturbations of size $\epsilon = 8/255$ (an improvement of +8.96% upon the state-of-the-art). Notably, our best CIFAR-10 models beat all techniques that use additional data (see Sec. 6) and constitutes one of the largest improvements ever made in the setting without additional data. As a consequence, we demonstrate that it is possible to avoid the use of 80M-TI [65] which has been withdrawn due to presence of offensive images.[1]

## 2   Related work

**Adversarial $\ell_p$ norm-bounded attacks.**   Since Biggio et al. [4], Szegedy et al. [64] observed that neural networks which achieve high accuracy are highly vulnerable to adversarial examples, the art of crafting increasingly sophisticated adversarial examples has received a lot of attention. Goodfellow et al. [27] proposed the Fast Gradient Sign Method (FGSM) which generates adversarial examples

---

[1]https://groups.csail.mit.edu/vision/TinyImages/

with a single normalized gradient step. It was followed by R+FGSM [67], which adds a randomization step, and the Basic Iterative Method (BIM) [44], which takes multiple smaller gradient steps.

**Adversarial training as a defense.** Adversarial training [49] is widely regarded as one of the most successful methods to train deep neural networks robust to such attacks. It has received significant attention and various modifications have emerged [25, 50, 76]. A notable work is TRADES [82], which balances the trade-off between standard and robust accuracy, and achieved state-of-the-art performance against $\ell_\infty$ norm-bounded perturbations on CIFAR-10. More recently, the work from Rice et al. [60] studied *robust overfitting* and demonstrated that improvements similar to TRADES could be obtained more easily using classical adversarial training with early stopping. Finally, Gowal et al. [30] highlighted how different hyper-parameters (such as network size and model weight averaging) affect robustness.

**Data-driven augmentations.** Works, such as *AutoAugment* [18] and related *RandAugment* [19], learn augmentation policies directly from data. These methods are tuned to improve standard classification accuracy and have been shown to work well on multiple datasets. *DeepAugment* [34] explores how perturbations of the parameters of pre-trained image-to-image models can be used to generate augmented datasets that provide increased robustness to common corruptions [32]. Similarly, generative models can be used to create novel views of images [37, 39, 57] by manipulating them in latent space. When optimized and used during training, these novel views reduce the impact of spurious correlations and improve accuracy [28, 73]. Most recently, Laidlaw et al. [45] proposed an adversarial training method based on bounding a neural perceptual distance (i.e., an approximation of the true perceptual distance). While these works make significant contributions towards improving generalization and robustness to semantic perturbations, they do not improve robustness to $\ell_p$ norm-bounded perturbations.

**Robustness to $\ell_p$ norm-bounded perturbations using generative modeling.** Finally, we highlight works, such as *Defense-GAN* [61] or *ME-Net* [77], which leverage data modeling techniques to create stronger defenses against $\ell_p$ norm-bounded attacks. Unfortunately, these techniques are not as robust as they seem and are broken by adaptive attacks [2, 16, 66]. Overall, to the best of our knowledge, there is little [48] to no evidence that data augmentations or generative models can be used to improve robustness to $\ell_p$ norm-bounded attacks. In fact, generative models mostly lack diversity and it is widely believed that the samples they produce cannot be used to train classifiers to the same accuracy than those trained on original datasets [59]. We differentiate ourselves from earlier works by leveraging additional generated samples for training rather than modifying the defense procedure, and by establishing sufficient conditions under which such samples improve robustness.

## 3 Adversarial training using generated data

The rest of this manuscript is organized as follows. In this section, we provide an overview of adversarial training, demonstrate using a motivational example that low-quality generated data can be leveraged to improve robustness to adversarial examples, and describe our method. In Sec. 4, we detail sufficient conditions that explain why generated samples can improve robustness and explore the limitations of our approach. In Sec. 5, we analyze four complementary and recent generative models in the context of our method. Finally, we provide experimental results in Sec. 6.

### 3.1 Adversarial training

For classification tasks, Madry et al. [49] propose to find model parameters $\boldsymbol{\theta}$ that minimize the adversarial risk:

$$\arg\min_{\boldsymbol{\theta}} \mathbb{E}_{(\boldsymbol{x},y)\sim\mathcal{D}} \left( \max_{\boldsymbol{\delta}\in\mathcal{S}} [f(\boldsymbol{x}+\boldsymbol{\delta};\boldsymbol{\theta}) \neq y] \right) \tag{1}$$

where $\mathcal{D}$ is a data distribution over pairs of examples $\boldsymbol{x}$ and corresponding labels $y$, $f(\cdot;\boldsymbol{\theta})$ is a model parametrized by $\boldsymbol{\theta}$, $[\cdot]$ is the Iverson bracket notation and corresponds to the $0-1$ loss, and $\mathcal{S}$ defines the set of allowed perturbations. For $\ell_p$ norm-bounded perturbations of size $\epsilon$, the perturbation set is defined as $\mathcal{S}_p = \{\boldsymbol{\delta} \mid \|\boldsymbol{\delta}\|_p \leq \epsilon\}$. Hence, for $\ell_\infty$ norm-bounded perturbations $\mathcal{S} = \mathcal{S}_\infty$ and for $\ell_2$ norm-bounded perturbations $\mathcal{S} = \mathcal{S}_2$. In the rest of this manuscript, we use $\epsilon_p$ to denote $\ell_p$ norm-bounded perturbations of size $\epsilon$ (e.g., $\epsilon_\infty = 8/255$).

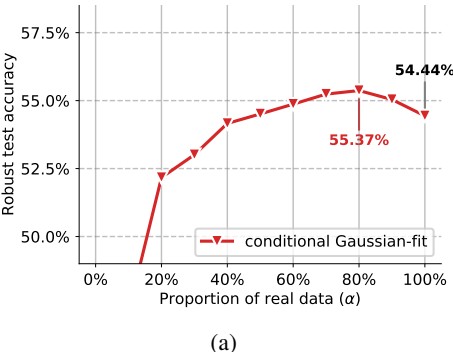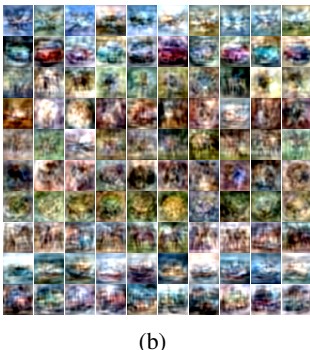

|  | |
|---|---|
| (a) | (b) |

Figure 3: Low-quality random inputs can improve robustness. Panel (a) shows the robust test accuracy (against AA+MT [30]) of a WRN-28-10 against $\epsilon_\infty = 8/255$ on CIFAR-10 trained with additional data randomly sampled from a class-conditional Gaussian fit of the training data. We compare how the proportion of original CIFAR-10 and generated images affects robustness (0% means generated samples only, while 100% means original CIFAR-10 train set only). Panel (b) shows some of the class-conditional Gaussian samples that are used during training.

In practice, given a training set $\mathcal{D}_{\text{train}}$, the adversarial training procedure replaces the $0-1$ loss with the cross-entropy loss $l_{\text{ce}}$ and is formulated as

$$\arg\min_{\boldsymbol{\theta}} \mathbb{E}_{(\boldsymbol{x},y)\in\mathcal{D}_{\text{train}}} \left( \max_{\boldsymbol{\delta}\in\mathcal{S}} l_{\text{ce}}(f(\boldsymbol{x}+\boldsymbol{\delta};\boldsymbol{\theta}),y) \right). \tag{2}$$

## 3.2 Generated data can improve robust generalization

Data augmentations can reduce the generalization error of standard (non-robust) training [18, 19, 23, 81]. However, to the contrary of standard training, augmentations beyond random flips and crops [31] – such as *Cutout* [23], *mixup* [81], *AutoAugment* [18] or *RandAugment* [19] – have been unsuccessful in the context of adversarial training [30, 60, 75]. The gap in robust accuracy between models trained with and without additional data suggests that common augmentation techniques, which tend to produce augmented views that are close to the original image they augment, are intrinsically limited in their ability to improve robust generalization. In other words, augmented samples are diverse (if the training set is diverse), but not complementary to the training set. This phenomenon is particularly exacerbated when training adversarially robust models which are known to require an amount of data polynomial in the number of input dimensions [62].

We hypothesize that, to improve robust generalization, it is critical to use additional training samples (augmented or generated) that are diverse and that complement the original training set (in the sense that these new samples should ideally come from the same underlying distribution as the training set but should not duplicate the training set). To test this hypothesis, we propose to use samples generated from a simple class-conditional Gaussian fit of the training data. By construction, such samples (shown in Fig. 3(b)) are extremely blurry but diverse. We proceed by fitting a multivariate Gaussian to each set of 5K training images corresponding to each class in CIFAR-10. For each class, we sample 100K images resulting in a new dataset of 1M datapoints (no further filtering is applied). In Fig. 3(a), we show the performance of various robust models trained by decreasing the proportion of real samples present in each batch from 100% (original data only) to 0% (generated data only). Decreasing this proportion reduces the importance of the original data. We observe that all proportions between 50% and 90% provide improvements in robust accuracy. Most surprisingly, the optimal proportion of 80% provides an absolute improvement of +0.93%, which is an improvement comparable in size to the ones provided by model weight averaging or TRADES [30]. As we show in Sec. 4, the drop in robust accuracy for proportions below 50% is expected in the capacity-limited regime. This experiment directly motivates our method.

## 3.3 Method

Given access to a pre-trained non-robust classifier $f_{\text{NR}}$ and an unconditional generative model approximating the true data distribution $\mathcal{D}$ by a distribution $\hat{\mathcal{D}}$, we would like to train a robust classifier $f(\cdot; \boldsymbol{\theta})$ parameterized by $\boldsymbol{\theta}$. We propose the following optimization problem:

$$\arg\min_{\boldsymbol{\theta}} \alpha \cdot \mathop{\mathbb{E}}_{(\boldsymbol{x},y)\in\mathcal{D}_{\text{train}}} \left( \max_{\boldsymbol{\delta}\in\mathcal{S}} l_{\text{ce}}(f(\boldsymbol{x}+\boldsymbol{\delta};\boldsymbol{\theta}),y) \right) + (1-\alpha) \cdot \mathop{\mathbb{E}}_{\boldsymbol{x}\sim\hat{\mathcal{D}}} \left( \max_{\boldsymbol{\delta}\in\mathcal{S}} l_{\text{ce}}(f(\boldsymbol{x}+\boldsymbol{\delta};\boldsymbol{\theta}),f_{\text{NR}}(\boldsymbol{x})) \right)$$

(3)

where $\alpha$ corresponds to a mixing factor that blends examples from the training set with those that are generated. When $\alpha$ is set to one, our method reverts back to the original adversarial training formulation in Eq. 2. When $\alpha$ is set to zero, our method only uses generated samples with their corresponding pseudo-labels. In practice, for efficiency, rather than generating samples on-the-fly, we pre-generate samples offline. Hence, both the original training set $\mathcal{D}_{\text{train}}$ and generated set $\hat{\mathcal{D}}$ contain a finite number of samples. We have the advantage, however, to be able to generate significantly more samples than present in the original training set. In App. B, we evaluate how varying the number of generated samples impacts adversarial robustness.

Overall, the complete method is described in Fig. 2 and is composed of three steps: *(i)* it starts by training the non-robust classifier and generative model on the original training set (for CIFAR-10, that corresponds to 50K images only); *(ii)* then, the generated dataset is constructed by drawing samples from the generative model and pseudo-labeling them using the non-robust classifier; *(iii)* finally, the robust classifier is trained using both the original training set and the generated dataset using Eq. 3.

## 4 Randomness might be enough

In this section, we formalize our notation (Sec. 4.1), and provide three sufficient conditions that explain why generated data can improve robustness (Sec. 4.2). In summary, *(i)* the pre-trained, non-robust classifier $f_{\text{NR}}$ used for pseudo-labeling must be accurate, *(ii)* the likelihood of sampling examples that are adversarial to this non-robust classifier must be low, and *(iii)* the generative model must be able to sample images from the true data distribution with non-zero probability.

### 4.1 Setup

Given a ground-truth function $f^\star$, we would like to find optimal parameters $\boldsymbol{\theta}^\star$ for $f(\cdot; \boldsymbol{\theta}^\star)$ that minimize the adversarial risk,

$$\boldsymbol{\theta}^\star = \arg\min_{\boldsymbol{\theta}} \mathbb{E}_{\boldsymbol{x}\sim\mathcal{D}} \left( \max_{\boldsymbol{\delta}\in\mathcal{S}} [f(\boldsymbol{x}+\boldsymbol{\delta};\boldsymbol{\theta}) \neq f^\star(\boldsymbol{x})] \right), \tag{4}$$

without access to the true data distribution $\mathcal{D}$ or the ground-truth classifier $f^\star$. As such, we replace the distribution $\mathcal{D}$ with an approximated distribution $\hat{\mathcal{D}}$ (from a generative model) and use a pre-trained non-robust classifier $f_{\text{NR}}$ instead of $f^\star$ (see Sec. 3.3). This results in sub-optimal parameters

$$\hat{\boldsymbol{\theta}}^\star = \arg\min_{\boldsymbol{\theta}} \mathbb{E}_{\boldsymbol{x}\sim\hat{\mathcal{D}}} \left( \max_{\boldsymbol{\delta}\in\mathcal{S}} [f(\boldsymbol{x}+\boldsymbol{\delta};\boldsymbol{\theta}) \neq f_{\text{NR}}(\boldsymbol{x})] \right). \tag{5}$$

We introduce the unknown probability measure $\mu$ corresponding to the true data distribution $\mathcal{D}$ and defined over the set of inputs $\mathcal{A} \subseteq \mathbb{R}^n$ (where $n$ is the input dimensionality), as well as the known probability measure $\hat{\mu}$ corresponding to the approximated distribution $\hat{\mathcal{D}}$. The set of relevant inputs $\mathcal{X} \subseteq \mathcal{A}$ (i.e., the set of *realistic* images for which we would like to enforce robustness) is the support of $\mu$ such that $\mu(\mathcal{X}) = 1$ and $\forall \mathcal{W} \subseteq \mathcal{X}, \mu(\mathcal{W}) > 0$ if $\mathcal{W}$ is non-empty. We assume that each input $\boldsymbol{x} \in \mathcal{X}$ can be assigned a label $y = f^\star(\boldsymbol{x})$ where $f^\star : \mathcal{X} \mapsto \mathcal{Y}$ is the ground-truth classifier (only valid for *realistic* images) and $\mathcal{Y} \in 2^{\mathbb{Z}}$ is the set of labels. Finally, given a perturbation set $\mathcal{S}$, we restrict labels such that there exists no *realistic* image within the perturbation set of another that has a different label; i.e., for $\boldsymbol{x} \in \mathcal{X}$, for all $\boldsymbol{\delta} \in \{\boldsymbol{\delta}' \in \mathcal{S} | \boldsymbol{x} + \boldsymbol{\delta}' \in \mathcal{X}\}$ we have $f^\star(\boldsymbol{x}) = f^\star(\boldsymbol{x} + \boldsymbol{\delta})$.

### 4.2 Limitations and sufficient conditions

To understand the limitations of our approach, it is useful to think about idealized sufficient conditions that would allow the sub-optimal parameters $\hat{\boldsymbol{\theta}}^\star$ to approach the performance of the optimal

parameters $\boldsymbol{\theta}^\star$. First, we concentrate on the capacity-limited regime and later extrapolate to the infinite-capacity, infinite-compute regime to gain more insights. The first sufficient condition concerns the pre-trained non-robust classifier and holds for both regimes.

**Condition 1** (accurate non-robust classifier). *The pre-trained non-robust classifier $f_{NR} : \mathcal{A} \mapsto \mathcal{Y}$ must be accurate on all* realistic *inputs $\boldsymbol{x} \in \mathcal{X}$: $\forall \boldsymbol{x} \in \mathcal{X}, f_{NR}(x) = f^\star(x)$.*

Indeed, if we had access to the true distribution $\mathcal{D}$, Eq. 4 and Eq. 5 could be made equal by setting $f^\star(x) = f_{\text{NR}}(x)$. [2] In the capacity-limited regime, when Cond. 1 is satisfied, the problem reduces to a robust generalization problem. This problem is widely studied [3, 21, 70] and one can show that the adversarial risk is bounded by the Wasserstein distance between the training distribution $\hat{\mathcal{D}}$ and true data distribution $\mathcal{D}$ (under mild assumptions) [46]. In other words, as $\hat{\mathcal{D}}$ approaches $\mathcal{D}$, we expect the robust accuracy of $f(\cdot; \hat{\boldsymbol{\theta}}^\star)$ to approach the one of $f(\cdot; \boldsymbol{\theta}^\star)$. This intuitively leads to the second sufficient condition and Prop. 1.

**Condition 2** (accurate approximated distribution). *The approximated data distribution $\hat{\mathcal{D}}$ and true data distribution $\mathcal{D}$ must be equivalent: $\mu(\mathcal{W}) = \hat{\mu}(\mathcal{W})$ for all measurable subset $\mathcal{W} \subseteq \mathcal{X}$.*

**Proposition 1** (capacity-limited regime). *Cond. 1 and Cond. 2 are sufficient conditions that allow the sub-optimal parameters $\hat{\boldsymbol{\theta}}^\star$ to match the performance of the optimal parameters $\boldsymbol{\theta}^\star$.*

Together, Cond. 1 and 2 provide sufficient conditions for the capacity-limited regime (see proof in Sec. E.1). Cond. 2 (and associated bounds from [46]) generally indicates that the robust accuracy of the classifier $f(\cdot; \hat{\boldsymbol{\theta}}^\star)$ should increase as the quality of the generative model that provides the approximated distribution $\hat{\mathcal{D}}$ improves. However, these two conditions do not provide a satisfying answer when it comes to understanding why seemingly random data can help improve robustness (as demonstrated in Sec. 3.2). To help our understanding, it is worth analyzing the consequence of increasing the capacity of $f$. In particular, in the infinite-capacity regime, Cond. 2 can be relaxed and replaced by the following two conditions, and Prop. 1 becomes Prop. 2.

**Condition 3** (unlikely adversarial examples). *It is not possible to sample a point $\boldsymbol{x} \sim \hat{\mathcal{D}}$ outside the* realistic *set $\mathcal{X}$ such that it is adversarial to $f_{NR}$: $\hat{\mu}(\mathcal{W}) = 0$ on the measurable subset $\mathcal{W} = \{x + \delta \mid x \in \mathcal{X}, \delta \in \mathcal{S}, f_{NR}(x + \delta) \neq f_{NR}(x)\}$.*

**Condition 4** (sufficient coverage). *The likelihood of any finite sample in the set of* realistic *inputs $\mathcal{X}$ obtained from $\hat{\mathcal{D}}$ should be non-zero under the measure $\hat{\mu}$: $\hat{\mu}(\mathcal{W}) > 0$ for all open measurable subsets $\mathcal{W} \subseteq \mathcal{X}$.*

**Proposition 2** (infinite-capacity regime). *Cond. 1, Cond. 3 and Cond. 4 are sufficient conditions that allow the sub-optimal parameters $\hat{\boldsymbol{\theta}}^\star$ to match the performance of the optimal parameters $\boldsymbol{\theta}^\star$ when the model $f$ has infinite capacity.*

Cond. 3 enforces that labels are non-conflicting within the perturbation set of a *realistic* input,[3] while Cond. 4 guarantees that *realistic* inputs appear with enough frequency during training. Together, Cond. 1, 3 and 4 do not only provide sufficient conditions for the infinite-capacity regime (see proof in Sec. E.1), but also explain why samples generated by a simple class-conditional Gaussian-fit can be used to improve robustness. Indeed, they imply that it is not necessary to have access to either the true data distribution or a perfect generative model when given enough compute and capacity. However, when compute and capacity are limited, it is critical that the optimization in Eq. 5 focuses on *realistic* inputs and that the distribution $\hat{\mathcal{D}}$ be as close as possible to the true distribution $\mathcal{D}$. In practice, this translates to the fact that better generative models (such as DDPM) can be used to achieve better robustness. We have relegated a discussion about the theoretical impact of the mixing factor $\alpha$ in Sec. E.2. Briefly stated, increasing $\alpha$ improves the realism of training samples (since the training samples mostly come from the original training set), but comes at the cost of a reduction in complementarity with the training set.

---

[2]In Sec. 6, we show that sub-optimal parameters $\hat{\boldsymbol{\theta}}^\star$ that improve upon those obtained by Eq. 2 can be obtained even when the non-robust classifier $f_{\text{NR}}$ is not perfect. In our experiments, we use a classifier that achieves 95.68% on the CIFAR-10 test set.

[3]Unless the generative model is trained to produce adversarial examples, random sampling is unlikely to produce images that are adversarial [8]. In fact, even strong black-box adversarial attacks require thousands of model queries to find adversarial examples.

Table 1: Complementarity and coverage of augmented and generated samples. We sample 10K images from the train set and various different generative models. For each sample in each set, we find its closest neighbor in Inception feature space (obtained after the pooling layer). To estimate complementarity, we report the proportion of samples with a nearest neighbor in either the train set, test set or the sampled set itself. To estimate coverage, we report the proportion of unique neighbors in the train and test set. We also include the IS and FID computed from 50K samples from each set and the robust accuracy obtained by a WRN-28-10 models trained on 1M samples (Sec. 6).

| | COMPLEMENTARITY | | | COVERAGE | | INCEPTION METRICS | | ROBUST |
| SETUP | TRAIN | TEST | SELF | TRAIN | TEST | IS ↑ | FID ↓ | ACCURACY ↑ |
|---|---|---|---|---|---|---|---|---|
| *mixup* [81] | 90.34% | 3.91% | 5.75% | 90.43% | 45.61% | $9.33 \pm 0.22$ | 7.71 | |
| Class-conditional Gaussian-fit | 0.13% | 0.22% | 99.65% | 12.36% | 12.24% | $3.64 \pm 0.03$ | 117.62 | 55.37% |
| VDVAE [14] | 11.97% | 12.14% | 75.89% | 34.20% | 33.76% | $6.88 \pm 0.05$ | 26.44 | 55.51% |
| BigGAN [7] | 14.97% | 14.81% | 70.22% | 38.86% | 39.06% | $9.73 \pm 0.07$ | 13.78 | 55.99% |
| StyleGAN2 [40] | 28.13% | 27.22% | 44.65% | 50.16% | 48.29% | $10.04 \pm 0.11$ | 2.57 | 58.17% |
| DDPM [36] | 29.29% | 29.17% | 41.54% | 49.07% | 49.10% | $9.50 \pm 0.14$ | 3.15 | 60.73% |

# 5 Generative models

The derivations from Sec. 4 and the experiment performed in Sec. 3.2 strongly suggest that generative models, which are capable of creating novel images [54], are viable augmentation candidates for adversarial training.

**Generative models considered in this work.** In this work, we limit ourselves to generative models that are solely trained on the original train set, as we focus on how to improve robustness in the setting without external data. We consider four recent and fundamentally different models: *(i)* BigGAN [7]: one of the first large-scale application of Generative Adversarial Networks (GANs) which produced significant improvements in Frechet Inception Distance (FID) and Inception Score (IS) on CIFAR-10 (as well as on IMAGENET); *(ii)* VDVAE [14]: a hierarchical Variational AutoEncoder (VAE) which outperforms alternative VAE baselines; *(iii)* StyleGAN2 [40]: an improved version of StyleGAN which borrows interesting properties from the style transfer literature; and *(iv)* DDPM [36]: a diffusion probabilistic model based on Langevin dynamics that reaches state-of-the-art FID on CIFAR-10.[4] As we have done for the simpler class-conditional Gaussian-fit, for each model, we sample 100K images per class, resulting in 1M images in total (see App. D for details). Samples are shown in App. D.

**Analysis of complementary and coverage.** In Table 1, we evaluate how close Cond. 2 and Cond. 4 are to be satisfied in practice. To do so, we sample 10K images from each generative model. We also sample 10K images from the CIFAR-10 training set, and apply *mixup* to them as a point of comparison.[5] We observe that *mixup* achieves a similar IS to the BigGAN and DDPM models. In the left-most set of three columns, for each augmented or generated sample, we report whether its closest neighbor in the Inception[6] feature space belongs to the train set, test set or the generated set itself (more details are available in App. D). An ideal generative model should create samples that are equally likely to be close to images from each set. We observe that *mixup* tends to produce samples that are too close to the original train set and that lack complementarity, potentially explaining its limited usefulness in terms of improving adversarial robustness. Meanwhile, generated samples (including those from the class-conditional Gaussian-fit) are much more likely to be close to images of the test set. We also observe that the DDPM neighbor distribution matches more closely the ideal uniform distribution. Images generated by BigGAN and VDVAE tend to have their nearest neighbor among themselves which indicates that these samples are either far from the train and test distributions or produce overly similar samples. Images generated by StyleGAN2, which reach an FID of 2.57 and IS of 10.07 that are better than the DDPM scores, have a slightly worse neighbor distribution (indicating a slight memorization of the training set). The middle two columns measure the ratio of unique neighbors that are matched in the train and test set. This provides a rough approximation

---

[4]We use VDVAE, StyleGAN2 and DDPM checkpoints available online and train our own BigGAN.

[5]According to prior work, *mixup* is unable to improve robust accuracy beyond the one obtained with random cropping/flipping when using early stopping [60]. Table 5 in the appendix shows more data augmentations.

[6]Using the LPIPS [84] feature space (see Table 4 in the appendix) provides similar conclusions.

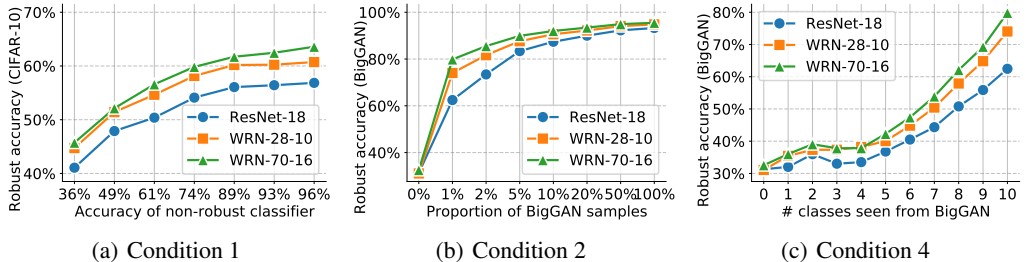

| (a) Condition 1 | (b) Condition 2 | (c) Condition 4 |

Figure 4: Impact of violations of the sufficient conditions detailed in Sec. 4. We report the robust test accuracy (against AA+MT [30]) when training different model architectures against $\epsilon_\infty = 8/255$. In panel (a), non-robust classifiers with different clean acccuracies are used for pseudo-labeling. In panel (b), we vary the mixture of training samples from a class-conditional Gaussian and a BigGAN distribution while the test distribution is the BigGAN distribution. In panel (c), we fix the proportion of samples from the class-conditional Gaussian to 99% and increase the number of classes from the BigGAN distribution seen during training (thus increasing coverage).

of coverage. We observe a similar trend where samples from the DDPM seem to provide a better coverage of the true data distribution. Note that, while these numbers rely on an inaccurate distance measure (i.e., Euclidean distance in Inception feature space) and should be taken with a grain of salt, they correlate well with the results obtained from our experiments. For example, models trained with StyleGAN2 samples obtain a lower robust accuracy than those trained with DDPM samples – despite obtaining better FID and IS.

## 6 Experiments

The experimental setup is explained in App. A. We use Residual Networks (ResNets) and Wide ResNets (WRNs) [31, 79] with Swish/SiLU [33] activations. We use stochastic weight averaging [38] with a decay rate of $0.995$. For adversarial training, we use TRADES [82] with 10 Projected Gradient Descent (PGD) steps. We train for $400$ CIFAR-10-equivalent epochs with a batch size of $1024$ (i.e., 19K steps). We evaluate our models against AUTOATTACK [16] and MULTITARGETED [29], which is denoted AA+MT [30]. For comparison, we trained ten WRN-28-10 models on CIFAR-10 (without additional generated samples) against $\epsilon_\infty = 8/255$. The resulting robust accuracy is $54.44\pm0.39\%$, thus showing a relatively low variance. Furthermore, as we will see, our best models are well clear of the threshold for statistical significance. On CIFAR-10 against $\epsilon_\infty = 8/255$ without additional generated samples a ResNet-18 achieves a robust accuracy of $50.64\%$ and a WRN-70-16 achieves $57.14\%$. Unless stated otherwise, all results pertain to CIFAR-10.

### 6.1 Sufficient conditions

The first set of experiments probes how violations of Cond. 1, 2 and 4 impact robustness against $\epsilon_\infty = 8/255$ (violations to Cond. 3 have an impact equivalent to those of Cond. 1). All experiments are summarized in Fig. 4 where we train models with increasing capacity.

**Non-robust classifier accuracy.** In Fig. 4(a), we train models using 1M samples generated by the DDPM and vary the accuracy of the the pre-trained, non-robust classifier $f_{NR}$. We evaluate the robust accuracy obtained on CIFAR-10 test set. We observe that robustness improves as the accuracy of $f_{NR}$ increases. Notably, even with the $74.47\%$-accurate non-robust classifier, the WRN-28-10 and WRN-70-16 obtain robust accuracies of $58.15\%$ and $59.83\%$, respectively, and already improve upon the state-of-the-art ($57.14\%$ at the time of writing). Thus, validating that, in practice, it is not necessary to have access to a perfect non-robust classifier.

**Quality of the generative models.** To analyze how the quality of the generative model influences robustness, we use the BigGAN to model the "true" data distribution. During training, we use samples generated from a mixture of the class-conditional Gaussian and BigGAN distributions; during testing, we evaluate on a separate subset of 10K unseen BigGAN samples. To probe Cond. 2, we

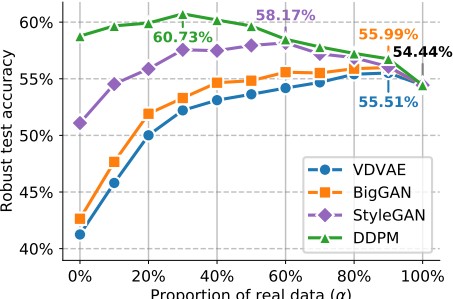

Figure 5: Robust test accuracy obtained by training a WRN-28-10 against $\epsilon_\infty = 8/255$ on CIFAR-10 when using additional data produced by different generative models. We compare how the ratio between original and generated images (i.e., $\alpha$) affects robustness (0% means generated samples only, 100% means CIFAR-10 train set only).

Table 2: Clean (without perturbations) and robust (under adversarial attack) accuracy obtained by different models (we pick the worst accuracy obtained by either AUTOATTACK or AA+MT). The accuracies are reported on the full test sets. For CIFAR-10, we test against $\epsilon_\infty = 8/255$ and $\epsilon_2 = 128/255$. For CIFAR-100, SVHN and TINYIMAGENET, we test against $\epsilon_\infty = 8/255$. * This model is trained for 2000 epochs on 100M samples.

| MODEL | DATASET | NORM | CLEAN | ROBUST |
|---|---|---|---|---|
| Wu et al. [75] (WRN-34-10) | CIFAR-10 | $\ell_\infty$ | 85.36% | 56.17% |
| Gowal et al. [30] (WRN-70-16) | | | 85.29% | 57.14% |
| Ours (DDPM) (WRN-28-10) | | | 85.97% | 60.73% |
| Ours (DDPM) (WRN-70-16) | | | 86.94% | 63.58% |
| Ours (100M DDPM)* (ResNet-18) | | | 87.35% | 58.50% |
| Ours (100M DDPM)* (WRN-28-10) | | | 87.50% | 63.38% |
| Ours (100M DDPM)* (WRN-70-16) | | | **88.74%** | **66.10%** |
| Wu et al. [75] (WRN-34-10) | CIFAR-10 | $\ell_2$ | 88.51% | 73.66% |
| Gowal et al. [30] (WRN-70-16) | | | **90.90%** | 74.50% |
| Ours (DDPM) (WRN-28-10) | | | 90.24% | 77.37% |
| Ours (DDPM) (WRN-70-16) | | | 90.83% | **78.31%** |
| Cui et al. [20] (WRN-34-10) | CIFAR-100 | $\ell_\infty$ | 60.64% | 29.33% |
| Gowal et al. [30] (WRN-70-16) | | | **60.86%** | 30.03% |
| Ours (DDPM) (WRN-28-10) | | | 59.18% | 30.81% |
| Ours (DDPM) (WRN-70-16) | | | 60.46% | **33.49%** |
| Ours (without DDPM) (WRN-28-10) | SVHN | $\ell_\infty$ | 92.87% | 56.83% |
| Ours (DDPM) (WRN-28-10) | | | **94.15%** | **60.90%** |
| Ours (without DDPM) (WRN-28-10) | TINYIMAGENET | $\ell_\infty$ | 51.56% | 21.56% |
| Ours (DDPM) (WRN-28-10) | | | **60.95%** | **26.66%** |

change the proportion of training samples from the class-conditional Gaussian. In effect, decreasing the proportion of such samples skews the mixed generative model (modeled by the mixture of Gaussian and BigGAN distributions) to produce more samples from the true distribution (modeled by the BigGAN distribution), thereby closing the gap between the approximated distribution $\hat{\mathcal{D}}$ and true distribution $\mathcal{D}$. As expected, Fig. 4(b) demonstrates that, given enough capacity, models can significantly reduce the adversarial risk.

**Relationship between coverage and capacity.** Similarly to Fig. 4(b), we use samples generated from a mixture of the class-conditional Gaussian and BigGAN distributions; during testing, we evaluate on a separate subset of 10K unseen BigGAN samples. To probe Cond. 4, we keep the proportion of samples from the class-conditional Gaussian distribution fixed at 99% and use the remaining 1% to include BigGAN samples corresponding to either 0, 1, … or 10 classes (thereby increasing coverage). In other words, the coverage of the true data distribution $\mathcal{D}$ (given by the BigGAN) increases as the number of seen classes increases. However, the approximated distribution $\hat{\mathcal{D}}$ remains different from the true data distribution even when the coverage reaches all classes (as the proportion of Gaussian samples is fixed to 99%). We observe in Fig. 4(c) that the robust accuracy of models with lower capacity improves less drastically – yielding a gap of 17.37% at full coverage between the ResNet-18 and WRN-70-16 models. This observation confirms that, with enough coverage, model capacity can compensate for the lack of a perfect generative model.

**Discussion.** Overall, Fig. 4(b) shows that when Cond. 2 is satisfied, the difference between models reduces and capacity takes a secondary role (since all models can bring their adversarial risk close to zero). Fig. 4(c) shows that when Cond. 4 is satisfied (and Cond. 2 is not), capacity matters as we observe that larger models benefit more from increased coverage. Both figures point to the fact that the quality of the generative model becomes less important when the capacity of the classifiers increases (as long as coverage is sufficient).

## 6.2 State-of-the-art robust accuracy

**Effect of mixing factor ($\alpha$).** As done in Sec. 3.3, we vary the proportion $\alpha$ of original images in each batch for all generated datasets. Fig. 5 explores a wide range of proportions while training a

WRN-28-10 against $\epsilon_\infty = 8/255$ on CIFAR-10. Samples from all models improve robustness when mixed optimally, but only samples from the StyleGAN2 and DDPM improve robustness significantly (+3.73% and +6.29%, respectively). It is also interesting to observe that, in the case of the DDPM, using 1M generated images is better than using the 50K images from the original train set only. While this may seem surprising, it can easily be explained if we assume that the DDPM produces many more high-quality, high-diversity images than the limited set of images present in the original data (c.f. [62]). We also observe that the optimal mixing factor is different for different generative models. Indeed, increasing $\alpha$ reduces the gap to the true data distribution at the cost of less complementarity with the original train set (see Sec. E.2).

**CIFAR-10.** Table 2 shows the performance of models trained with 1M samples generated by the DDPM on CIFAR-10 against $\epsilon_\infty = 8/255$ and $\epsilon_2 = 128/255$. Irrespective of their size, models trained with 1M DDPM samples surpass the current state-of-the-art in robust accuracy by a large margin (+6.44% and +3.81%). When using 100M DDPM samples (and training for 2000 epochs), we reach 66.10% robust accuracy against $\epsilon_\infty = 8/255$ which constitutes an improvement of +8.96% over the state-of-the-art. In this setting, our smallest model (ResNet-18) surpasses state-of-the-art results obtained by much larger models (e.g., WRN-70-16). Most remarkably, despite not using any external data, against $\epsilon_\infty = 8/255$, our best model beats all RobustBench [17] entries that used external data (see Table 6 in the appendix).

**Generalization to other datasets (CIFAR-100, SVHN and TINYIMAGENET).** Finally, to evaluate the generality of our approach, we evaluate it on CIFAR-100, SVHN [53] and TINYIMAGENET [26]. We train two new DDPM on the train set of CIFAR-100 and SVHN and sample 1M images from each. For TINYIMAGENET, we use a DDPM trained on IMAGENET [24] at $64 \times 64$ resolution and restricted samples to the 200 classes of TINYIMAGENET. The results are shown in Table 2. On CIFAR-100, our best model reaches a robust accuracy of 33.49% and improves noticeably upon the state-of-the-art by +3.46% (in the setting that does not use any external data). On SVHN, in the same table, we compare models trained without and with DDPM samples. Again, the addition of DDPM samples significantly improves robustness, with the robust accuracy improving by by +4.07%. On TINYIMAGENET, the improvement is +5.10%.

## 7 Conclusion.

Using generative models, we posit and demonstrate that generated samples provide a greater diversity of augmentations that allow adversarial training to go well beyond the current state-of-the-art. Our work provides novel insights into the effect of diversity and complementarity on robustness, which we hope can further our understanding of robustness. All our models and generated datasets are available online at https://github.com/deepmind/deepmind-research/tree/master/adversarial_robustness.

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
