# A  Experimental setup

The implementation of the following setup is written in JAX [6] and Haiku [35].

**Architecture.** We use Residual Networks (ResNets) and Wide ResNets (WRNs) [31, 79]. This is consistent with prior work [30, 49, 60, 72, 82] which use diverse variants of these network families. Furthermore, we adopt the same architecture details as Gowal et al. [30] with Swish/SiLU [33] activation functions. Most of the experiments are conducted on a WRN-28-10 model which has a depth of 28, a width multiplier of 10 and contains 36M parameters. To evaluate the effect of using additional generated data on wider and deeper networks, we also run several experiments using WRN-70-16, which contains 267M parameters.

**Outer minimization.** We use TRADES [82] optimized using SGD with Nesterov momentum [52, 58] and a global weight decay of $5 \times 10^{-4}$. We use a batch size to $1024$ split over 32 Google Cloud TPUv3 cores [43], train for $400$ CIFAR-10-equivalent epochs (resulting in 19K training steps), and use a *cosine* learning rate schedule [47] without restarts where the initial learning rate is set to 0.4 and is decayed to 0 by the end of training (similar to [30]). We also use model weight averaging (WA) [38] with a decay rate of $\tau = 0.995$. With this setup, training a WRN-28-10, a WRN-70-16 and a ResNet-18 takes 2.5 hours, 6 hours and 22 minutes, respectively.

**Inner minimization.** Adversarial examples are obtained by maximizing the Kullback-Leibler divergence between the predictions made on clean inputs and those made on adversarial inputs [82]. This optimization procedure is done using the Adam optimizer [41] with a step-size of $0.1$ and $10$ steps.

**Evaluation.** We follow the evaluation protocol designed by Gowal et al. [30]. Specifically, we train two (and only two) models for each hyperparameter setting, perform early stopping for each model on a separate validation set of 1024 samples using PGD$^{40}$ (i.e., PGD with 40 gradient ascent steps) similarly to Rice et al. [60] and pick the best model by evaluating the robust accuracy on the same validation set. The average absolute difference between these two models is -0.12% in test robust accuracy (as measured over 10 separate runs). Unless stated otherwise, we always report the robust test accuracy against a mixture of AUTOATTACK [16] and MULTITARGETED [29], which is denoted by AA+MT. This mixture consists in completing the following sequence of attacks: AUTOPGD on the cross-entropy loss with 5 restarts and 100 steps, AUTOPGD on the difference of logits ratio loss with 5 restarts and 100 steps and finally MULTITARGETED on the margin loss with 10 restarts and 200 steps. We note that, while early stopping is not necessary when using the cosine learning rate schedule, we keep it to be consistent with prior work.

# B  Additional results

**Scaling dataset size.** Using a generative model allows us to sample many more images than available in the original training set. In Fig. 6, we set the mixing factor $\alpha$ to zero (thus only using generated samples) and vary the number of training samples. We evaluate the robust accuracy of the resulting model on the CIFAR-10 test set and on a separate validation set composed of 10K generated samples. We also compare models trained on BigGAN and DDPM samples. Irrespective of the underlying generative model, using more samples generally improves robustness. Samples from the

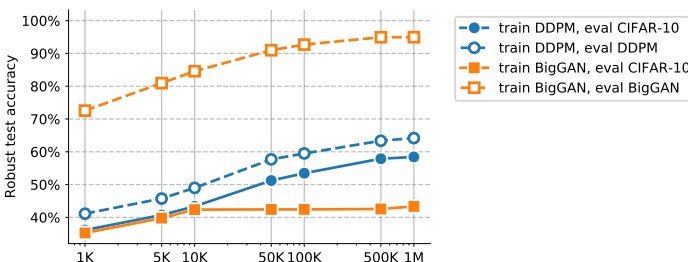

Figure 6: Robust test accuracy when training a WRN-28-10 using a variable number of samples from a DDPM or BigGAN. We compare the robust accuracy on the CIFAR-10 test set with the one obtained on a separate set of generated samples.

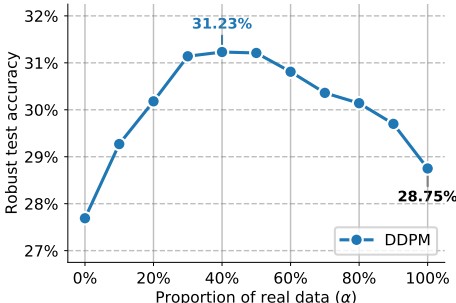

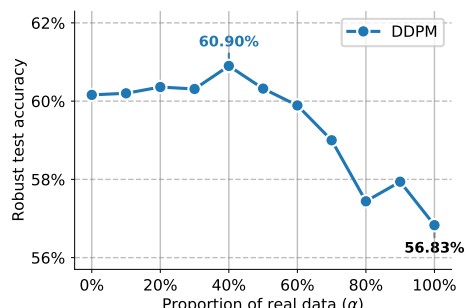

Figure 7: Robust test accuracy when training a WRN-28-10 against $\epsilon_\infty = 8/255$ on CIFAR-100 with additional data produced by a DDPM. We compare how the ratio between original images and generated images in the training minibatches affects the test robust performance (0% means generated samples only, while 100% means original CIFAR-100 train set only).

Figure 8: Robust test accuracy when training a WRN-28-10 against $\epsilon_\infty = 8/255$ on SVHN with additional data produced by a DDPM. We compare how the ratio between original images and generated images in the training minibatches affects the test robust performance (0% means generated samples only, while 100% means original SVHN train set only).

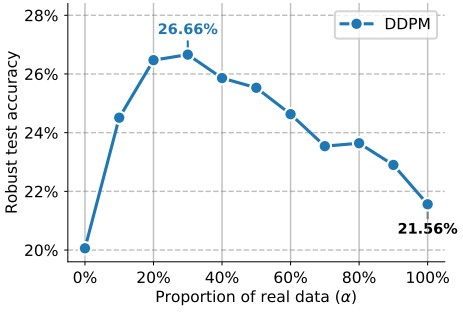

Figure 9: Robust test accuracy when training a WRN-28-10 against $\epsilon_\infty = 8/255$ on TINY-IMAGENET with additional data produced by a DDPM trained on IMAGENET. We compare how the ratio between original images and generated images in the training minibatches affects the test robust performance (0% means generated samples only, while 100% means original TINYIMAGENET train set only).

DDPM are more useful, as can be seen from the higher robust accuracy obtained on the CIFAR-10 test set (i.e., 58.43% versus 43.34% with a WRN-28-10). It is also worth noting that using samples from the DDPM results in a smaller generalization gap of 5.74 points when using 1M samples (gap between the dashed and solid blue lines). Models trained on BigGAN samples tend to overfit to these samples, which results in a large generalization gap of 51.61 points (gap between the dashed and solid orange lines). These results also confirm that BigGAN samples are easier to robustly classify (possibly due to their low diversity).

**CIFAR-100.** For completeness, we also report the effect of mixing different proportions of generated and original samples in Fig. 7 against $\epsilon_\infty = 8/255$ using a WRN-28-10 on CIFAR-100. Similarly to Fig. 5, we observe that additional samples generated by DDPM are useful to improve robustness, with an absolute improvement of +2.48% in robust accuracy.

**SVHN.** We report the effect of mixing different proportions of generated and original samples in Fig. 8 against $\epsilon_\infty = 8/255$ using a WRN-28-10 on SVHN. Similarly to Fig. 5 and Fig. 7, we observe that additional samples generated by DDPM are useful to improve robustness, with an absolute improvement of +4.07% in robust accuracy.

**TINYIMAGENET.** We report the effect of mixing different proportions of generated and original samples in Fig. 9 against $\epsilon_\infty = 8/255$ using a WRN-28-10 on TINYIMAGENET. Similarly to Fig. 5, Fig. 7 and Fig. 8, we observe that additional samples generated by DDPM are useful to improve robustness, with an absolute improvement of +5.10% in robust accuracy.

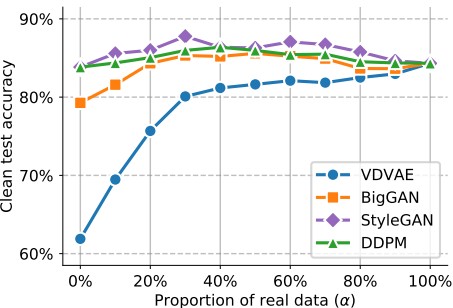

Figure 10: Clean test accuracy obtained by training a WRN-28-10 against $\epsilon_\infty = 8/255$ on CIFAR-10 when using additional data produced by different generative models. We compare how the ratio between original and generated images (i.e., $\alpha$) affects the clean accuracy (0% means generated samples only, 100% means CIFAR-10 train set only). The robust test accuracy for the same models is shown in Fig. 5 in the main manuscript.

**Clean accuracy.** Finally, the clean accuracy (i.e., accuracy obtained when no perturbation is applied to the input) of all models used in Fig. 5 is reported in Fig. 10. All these models are trained adversarially to be robust against $\epsilon_\infty = 8/255$ on CIFAR-10. We observe that improvements in robust accuracy are not always correlated (either positively or negatively) with improvements in clean accuracy. While VDVAE samples provide no improvements in clean accuracy, using BigGAN, StyleGAN2 or DDPM samples can improve clean accuracy by up to +1.27%, +3.45% and +2.05%, respectively.

## C    Analysis of models

In this section, we perform additional diagnostics that give us confidence that our models are not doing any form of gradient obfuscation or masking [2, 71].

**AUTOATTACK and robustness against black-box attacks.** First, we report in Table 3 the robust accuracy obtained by our strongest models against a diverse set of attacks. These attacks are run as a cascade using the AUTOATTACK library available at https://github.com/fra31/auto-attack. First, we observe that our combination of attacks, denoted AA+MT matches the final robust accuracy measured by AUTOATTACK. Second, we also notice that the black-box attack (i.e., SQUARE) does not find any additional adversarial examples. Overall, these results indicate that our empirical measurement of robustness is meaningful and that our models do not obfuscate gradients.

Table 3: Clean (without adversarial attacks) accuracy and robust accuracy (against the different stages of AUTOATTACK) on CIFAR-10 obtained by different models. Refer to https://github.com/fra31/auto-attack for more details.

| MODEL | DATASET | NORM | RADIUS | AUTOPGD-CE | + AUTOPGD-T | + FAB-T | + SQUARE | CLEAN | AA+MT |
|---|---|---|---|---|---|---|---|---|---|
| WRN-28-10 (DDPM) | CIFAR-10 | $\ell_\infty$ | $\epsilon = 8/255$ | 63.53% | 60.73% | 60.73% | 60.73% | 85.97% | 60.73% |
| WRN-70-16 (DDPM) | | | | 65.95% | 63.62% | 63.62% | 63.62% | 86.94% | 63.58% |
| ResNet-18 (100M DDPM) | | | | 60.85% | 58.63% | 58.63% | 58.63% | 87.35% | 58.50% |
| WRN-28-10 (100M DDPM) | | | | 65.65% | 63.44% | 63.44% | 63.44% | 87.50% | 63.38% |
| WRN-70-16 (100M DDPM) | | | | 68.46% | 66.13% | 66.11% | 66.11% | 88.74% | 66.10% |
| WRN-28-10 (DDPM) | CIFAR-10 | $\ell_2$ | $\epsilon = 128/255$ | 78.13% | 77.44% | 77.44% | 77.44% | 90.24% | 77.37% |
| WRN-70-16 (DDPM) | | | | 78.97% | 78.39% | 78.39% | 78.39% | 90.93% | 78.31% |
| WRN-28-10 (DDPM) | CIFAR-100 | $\ell_\infty$ | $\epsilon = 8/255$ | 34.47% | 30.81% | 30.81% | 30.81% | 59.18% | 31.23% |
| WRN-70-16 (DDPM) | | | | 36.27% | 33.49% | 33.49% | 33.49% | 60.46% | 33.93% |

**Loss landscapes.** We analyze the adversarial loss landscapes of our best model trained on CIFAR-10 against $\epsilon_\infty = 8/255$ (a WRN-70-16). To generate a loss landscape, we vary the network input along the linear space defined by the worse perturbation found by PGD$^{40}$ ($u$ direction) and a random Rademacher direction ($v$ direction). The $u$ and $v$ axes represent the magnitude of the perturbation added in each of these directions respectively and the $z$ axis is the adversarial margin loss [10]: $z_y - \max_{i \neq y} z_i$ (i.e., a misclassification occurs when this value falls below zero). Fig. 11 shows the loss landscapes around the first 2 images of the CIFAR-10 test set for the aforementioned model. Both landscapes are smooth and do not exhibit patterns of gradient obfuscation. Overall, it is difficult to interpret these figures further, but they do complement the numerical analyses done so far.

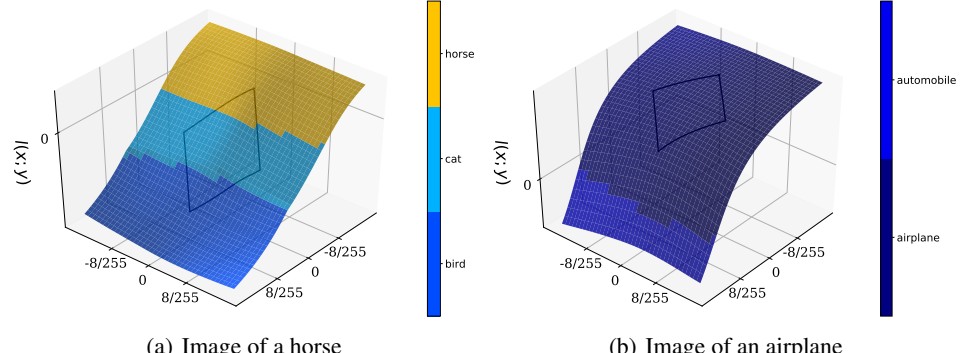

(a) Image of a horse

(b) Image of an airplane

Figure 11: Loss landscapes around the first two images from the CIFAR-10 test set for the WRN-70-16 networks trained with DDPM samples. It is generated by varying the input to the model, starting from the original input image toward either the worst attack found using $\text{PGD}^{40}$ ($u$ direction) or a random Rademacher direction ($v$ direction). The loss used for these plots is the margin loss $z_y - \max_{i \neq y} z_i$ (i.e., a misclassification occurs when this value falls below zero). The diamond-shape represents the projected $\ell_\infty$ ball of size $\epsilon = 8/255$ around the nominal image.

## D   Details on generated data

**Generative models.**   In this paper, we use four different and complementary generative models: *(i)* BigGAN [7], *(ii)* VDVAE [14], *(iii)* StyleGAN2 [40] and *(iv)* DDPM [36]. Except for BigGAN, we use the CIFAR-10 checkpoints that are available online. For BigGAN, we train our own model and pick the model that achieves the best FID (the model architecture and training schedule is the same as the one used in [7]). All models are trained solely on the CIFAR-10 train set (or the train set of CIFAR-100 or SVHN for experiments shown in App. B). As a baseline, we also fit a class-conditional multivariate Gaussian, which reaches FID and IS metrics of 117.62 and 3.64, respectively. We also report that BigGAN reaches an FID of 11.07 and IS of 9.71; VDVAE reaches an FID of 36.88 and IS of 6.03; StyleGAN2 reaches an FID of 2.57 and IS of 10.04 and DDPM reaches an FID of 3.15 and IS of 9.50.[7,8,9]

**Datasets of generated samples.**   For the class-conditional multivariate Gaussian and DDPM samples, we use a pretrained WRN-28-10 to give pseudo-labels. This WRN-28-10 is trained non-robustly on the CIFAR-10 train set and achieves 95.68% accuracy.[10,11,12] We sample images from each model until we have 100K images for each class.[13] For BigGAN and VDVAE samples, we proceed with an additional filtering step similarly to the one proposed by Carmon et al. [11]. We sample from each generative model 5M images and score each image using the pretrained, non-robust WRN-28-10 model used for pseudo-labeling. For each class, we select the top-100K scoring images and build a dataset of 1M image-label pairs.[14]

This additional generated data (consisting of 1M samples) is used to train adversarially robust models by mixing in each batch a given proportion of original and generated examples. Fig. 12 shows a random subset of this additional data for each generative model. We also report the FID and IS metrics of the resulting sets in Table 1 and Table 4. They might differ from metrics obtained by each

---

[7]For CIFAR-100, we trained our own DDPM which achieves an FID of 5.58 and IS of 10.82.

[8]For SVHN, we trained our own DDPM which achieves an FID of 4.89 and IS of 3.06.

[9]For TINYIMAGENET, we used the class-conditional DDPM checkpoint available at https://github.com/openai/guided-diffusion which has been trained on IMAGENET at a $64 \times 64$ resolution.

[10]For CIFAR-100, the same model achieves 79.98% accuracy.

[11]For SVHN, the same model achieves 96.54% accuracy.

[12]For TINYIMAGENET, we use a class-conditional StyleGAN2 model and do not need to train a non-robust classifier for pseudo-labeling.

[13]We use 10K images per class for CIFAR-100 experiments.

[14]All generated datasets are available online at https://github.com/deepmind/deepmind-research/tree/master/adversarial_robustness.

generative model individually (see previous paragraph) as we filter images to either keep the highest scoring ones or make sure that classes are balanced.

**Diversity and complementarity.** While the FID metric does capture how two distributions of samples match, it does not necessarily provide enough information in itself to assess the overlap between the distribution of generated samples and the train or test distributions (this is especially true for samples obtained through data augmentations such as *mixup*) – as seen in Table 1 and explained in the next paragraph. As such, we also decide to compute the proportion of nearest neighbors in perceptual space: given equal Inception metrics, a better generative model would produce samples that are equally likely to be close to training, testing or generated images. In an attempt to estimate the coverage of the real data distribution, we also compute the proportion of nearest neighbors that are unique: given equal Inception metrics, a better generative model would produce samples that are equally likely to be close to any image in train or test set (thus resulting a high proportion of unique neighbors).[15]

We now describe how we compute Table 1 which reports nearest-neighbors statistics for the different generative models. First, we sample 10K images from the train set of CIFAR-10 (uniformly across classes) and take the full test set of CIFAR-10. We then pass these 20K images through the pre-trained Inception network (used to measure Inception metrics). We use the activations from the last pooling operation and compute their top-100 PCA components, as this allows us to compare samples in a much lower dimensional space (i.e., 100 instead of 2048). Finally, for each generative model, we sample 10K images from their 1M dataset (class-balanced as well) and pass them through the pipeline composed of the Inception network and the PCA projection computed on the original data. The left-most three columns (entitled "complementarity") are computed by finding, for each generated sample, its closest neighbor in the PCA-reduced feature space to any image from the set of $30K - 1$ images composed of train, test and generated sets. We then measure whether this nearest-neighbor belongs to the original datasets of 10K image each (train or test) or to the generated set (self) composed of the remaining $10K - 1$ images. For example, given 6 generated samples (instead of the 10K), the first sample's closest neighbor could be in the train set, the next two samples' closest neighbors could in the test set and the last three samples' closest neighbors could be in the set of generated samples. This would result in ratios of $1/6$, $1/3$ and $1/2$. The middle set of two columns (entitled "coverage") is computed by finding, for each generated sample, its closest neighbor in the PCA-reduced feature space to any image from the train and test sets. We then measure the number of unique neighbors matched in both sets.[16] See Alg. 1 for pseudo-code.

---

**Algorithm 1** Complementarity and coverage computation

---

**Input:** Train set $\mathcal{D}_{\text{train}}$, test set $\mathcal{D}_{\text{test}}$, distribution $\hat{\mathcal{D}}$ for which we measure complementary and coverage, number of samples $N$ and a function $g : \mathbb{R}^n \mapsto \mathbb{R}^m$ that maps inputs to their features (e.g., Inception features).

**Output:** Complementarity $\{c_{\text{train}}, c_{\text{test}}, c_{\text{self}}\}$ and coverage $\{v_{\text{train}}, v_{\text{test}}\}$.

1: $\mathcal{D}_{\text{self}} \leftarrow \{\boldsymbol{x}_i \sim \hat{\mathcal{D}}\}_{i=1}^N$                                    ▷ Pick $N$ samples from $\hat{\mathcal{D}}$
2: $\bar{\mathcal{D}}_{\text{train}}$ is such that $\bar{\mathcal{D}}_{\text{train}} \subseteq \mathcal{D}_{\text{train}}$ and $|\bar{\mathcal{D}}_{\text{train}}| = N$          ▷ Pick $N$ samples from $\mathcal{D}_{\text{train}}$
3: $\bar{\mathcal{D}}_{\text{test}}$ is such that $\bar{\mathcal{D}}_{\text{test}} \subseteq \mathcal{D}_{\text{train}}$ and $|\bar{\mathcal{D}}_{\text{test}}| = N$            ▷ Pick $N$ samples from $\mathcal{D}_{\text{test}}$
4: $c_{\text{train}} \leftarrow 0, c_{\text{test}} \leftarrow 0, c_{\text{self}} \leftarrow 0$                      ▷ Initialize complementarity counters
5: $\mathcal{V}_{\text{train}} \leftarrow \emptyset, \mathcal{V}_{\text{test}} \leftarrow \emptyset$                                  ▷ Initialize coverage sets
6: **for** $\boldsymbol{x}_i \in \mathcal{D}_{\text{self}}$ **do**                                  ▷ For all generated samples
7:     $\bar{\mathcal{D}}_{\text{self}} = \mathcal{D}_{\text{self}} \setminus \{\boldsymbol{x}_i\}$                  ▷ Ignore current sample in computation below
8:     $s^\star = \arg\min_{s \in \{\text{train},\text{test},\text{self}\}} \min_{\boldsymbol{x}'_i \in \bar{\mathcal{D}}_s} \|g(\boldsymbol{x}_i) - g(\boldsymbol{x}'_i)\|_2$         ▷ Find closest set
9:     $c_{s^\star} \leftarrow c_{s^\star} + 1/N$                            ▷ Increment counter of closest set
10:     $\mathcal{V}_{\text{train}} \leftarrow \mathcal{V}_{\text{train}} \cup \{\arg\min_{\boldsymbol{x}'_i \in \bar{\mathcal{D}}_{\text{train}}} \|g(\boldsymbol{x}_i) - g(\boldsymbol{x}'_i)\|_2\}$   ▷ Find closest neighbor in train set
11:     $\mathcal{V}_{\text{test}} \leftarrow \mathcal{V}_{\text{test}} \cup \{\arg\min_{\boldsymbol{x}'_i \in \bar{\mathcal{D}}_{\text{test}}} \|g(\boldsymbol{x}_i) - g(\boldsymbol{x}'_i)\|_2\}$   ▷ Find closest neighbor in test set
12: **end for**
13: $v_{\text{train}} = |\mathcal{V}_{\text{train}}|/N, \, v_{\text{test}} = |\mathcal{V}_{\text{test}}|/N$                        ▷ Compute coverage ratio

---

[15] As a point of comparison, sampling 2 sets of 10K points from a uniform distribution $\mathcal{U}_{[0,1]}$ between 0 and 1 yields in average a proportion of unique nearest neighbors equal to 55.6%.

[16] According to this measure, the CIFAR-10 train set covers 52.27% of the test set, while the test set covers 52.08% of the train set. Obtaining a significantly higher coverage of the train set is likely the result of overfitting and memorization.

Table 4: Complementarity and coverage of augmented and generated samples. We sample 10K images from the train set and various different generative models. For each sample in each set, we find its closest neighbor in LPIPS feature space. To estimate complementarity, we report the proportion of samples with a nearest neighbor in either the train set, test set or the sampled set itself. To estimate coverage, we report the proportion of unique neighbors in the train and test set. We also include the IS and FID computed from 50K samples from each set.

| | COMPLEMENTARITY | | | COVERAGE | | INCEPTION METRICS | |
|---|---|---|---|---|---|---|---|
| SETUP | TRAIN | TEST | SELF | TRAIN | TEST | IS ↑ | FID ↓ |
| *mixup* [81] | 96.33% | 0.17% | 3.50% | 98.34% | 41.93% | $9.33 \pm 0.22$ | 7.71 |
| Class-conditional Gaussian-fit | 0.80% | 0.72% | 98.48% | 15.80% | 16.39% | $3.64 \pm 0.03$ | 117.62 |
| VDVAE [14] | 6.52% | 5.71% | 87.77% | 23.20% | 23.69% | $6.88 \pm 0.05$ | 26.44 |
| BigGAN [7] | 11.69% | 9.55% | 78.76% | 39.29% | 38.89% | $9.73 \pm 0.07$ | 13.78 |
| DDPM [36] | 31.20% | 26.39% | 42.41% | 44.08% | 43.80% | $9.50 \pm 0.14$ | 3.15 |

Table 5: Complementarity and coverage of augmented samples using the Inception feature space (as done in Table 1, but for additional data augmentation schemes).

| | COMPLEMENTARITY | | | COVERAGE | |
|---|---|---|---|---|---|
| SETUP | TRAIN | TEST | SELF | TRAIN | TEST |
| *mixup* [81] | 90.34% | 3.91% | 5.75% | 90.43% | 45.61% |
| *Cutout* [23] | 65.46% | 3.47% | 31.07% | 76.76% | 45.24% |
| *CutMix* [78] | 60.30% | 7.40% | 32.30% | 66.05% | 45.63% |
| *AutoAugment* [18] | 67.13% | 6.00% | 26.87% | 69.44% | 45.67% |
| *RandAugment* [19] | 61.23% | 8.85% | 29.92% | 65.51% | 45.78% |

In Table 4, we repeat the process used for Table 1 for a subset of its rows by using the pretrained VGG network which measures a Perceptual Image Patch Similarity, also known as LPIPS [84], instead of the Inception network. We use the resulting 124,928 concatenated activations and compute their top-100 PCA components. Overall, the resulting numbers are similar to the ones obtained by the Inception network. In Table 5, we use Inception features to compute the complementarity and coverage metrics of various data augmentation schemes. All augmentation schemes produce samples that are too close to the train set and too far from the test set, which indicates that when they provide samples that could complement the train set, these samples are far from the true distribution.

**Shortcomings of FID and IS.**   The coverage and complementary metrics from Table 1 provide additional information that is not captured by FID and IS. In particular, a generative model that memorizes the train set will produce almost perfect FID and IS scores, but will produce a neighbor distribution of 100% matching the train set, 0% matching the test set and 0% matching itself. This is far from the ideal distribution of $1/3, 1/3, 1/3$. Similarly, a generative model that focuses on a subset of the true distribution can produce high IS, but low coverage (as exemplified by BigGAN samples).

# E   Theoretical foundations

## E.1   Proofs

Let us consider Prop. 1 and Prop. 2 again.

**Proposition 1** (capacity-limited regime). *Cond. 1 and Cond. 2 are sufficient conditions that allow the sub-optimal parameters $\hat{\boldsymbol{\theta}}^{\star}$ to match the performance of the optimal parameters $\boldsymbol{\theta}^{\star}$.*

*Proof.* When $f_{\mathrm{NR}}(x) = f^{\star}(x)$ for all $\boldsymbol{x} \in \mathcal{X}$ and $\hat{\mathcal{D}}$ is such that $\mu(\mathcal{W}) = \hat{\mu}(\mathcal{W})$ for all measurable $\mathcal{W} \subseteq \mathcal{X}$, Eq. 4 and Eq. 5 become identical. Hence, their solutions achieve the same objective.  □

**Proposition 2** (infinite-capacity regime). *Cond. 1, Cond. 3 and Cond. 4 are sufficient conditions that allow the sub-optimal parameters $\hat{\boldsymbol{\theta}}^{\star}$ to match the performance of the optimal parameters $\boldsymbol{\theta}^{\star}$ when the model $f$ has infinite capacity.*

*Proof.* Cond. 3 guarantees that there are no images with conflicting labels within the perturbation set $\mathcal{S}$ of a *realistic* image (this extends the non-conflicting labels setup from Sec. 4.1) As such, it is possible to drive the objective from Eq. 5 to zero. Since $f_{\text{NR}}(x) = f^\star(x)$ for all $x \in \mathcal{X}$ (Cond. 1) and the generated data covers the true distribution (Cond. 4), the objective obtained from Eq. 5 can only be zero if the objective to Eq. 4 is also zero. Hence, the solutions of Eq. 4 and Eq. 5 achieve the same objective. $\qquad\square$

### E.2 Impact of the mixing factor $\alpha$

We address here the impact of the mixing factor $\alpha$ used in Eq. 3. Ignoring the change of loss, Eq. 3 can be formulated as Eq. 5 by using the non-robust classifier $f'_{\text{NR}}$ instead of $f_{\text{NR}}$ and using the merged distribution $\hat{\mathcal{D}}'$ instead of $\hat{\mathcal{D}}$, with

$$f'_{\text{NR}}(x) = \begin{cases} f^\star(\boldsymbol{x}) & \text{if } \boldsymbol{x} \in \mathcal{D}_{\text{train}} \\ f_{\text{NR}}(\boldsymbol{x}) & \text{otherwise} \end{cases} \qquad (6)$$

and with $\hat{\mathcal{D}}'$ such that sampling $\boldsymbol{x} \sim \hat{\mathcal{D}}'$ is equivalent to $\boldsymbol{x} = [r \leq \alpha]\boldsymbol{x}' + [r > \alpha]\boldsymbol{x}''$ with $r \sim \mathcal{U}_{[0,1]}, \boldsymbol{x}' \sim \mathcal{U}_{\mathcal{D}_{\text{train}}}$ and $\boldsymbol{x}'' \sim \hat{\mathcal{D}}$ (where $\mathcal{U}_{\mathbb{A}}$ corresponds to the uniform distribution over set $\mathbb{A}$). This transformation artificially improves the accuracy of the non-robust classifier and reduces the gap between $\mathcal{D}$ and $\hat{\mathcal{D}}'$, thus resulting in better coverage. Note, however, that while increasing $\alpha$ improves the realism of training samples, it comes at the cost of a reduction in complementarity with the training set.

## F  RobustBench

For reference, at the time of writing, the top-5 RobustBench (https://robustbench.github.io/ [17]) leaderboard entries without and with additional data are listed in Table 6. Entries from non-peer reviewed venues were only included if older than 2 months from writing.

Table 6: State of RobustBench leaderboard at the time of writing. We report the clean (without adversarial attacks) accuracy and robust accuracy on CIFAR-10 against $\epsilon_\infty = 8/255$.

| AUTHOR | MODEL | CLEAN | ROBUST |
|---|---|---|---|
| WITHOUT EXTERNAL DATA | | | |
| Gowal et al. [30] | WRN-70-16 | 85.29% | 57.14% |
| Gowal et al. [30] | WRN-34-20 | 85.64% | 56.82% |
| Wu et al. [75] | WRN-34-10 | 85.36% | 56.17% |
| Pang et al. [55] | WRN-34-20 | 86.43% | 54.39% |
| Pang et al. [56] | WRN-34-20 | 85.14% | 53.74% |
| WITH EXTERNAL DATA | | | |
| Gowal et al. [30] | WRN-70-16 | 91.10% | 65.87% |
| Gowal et al. [30] | WRN-34-20 | 89.48% | 62.76% |
| Wu et al. [74] | WRN-34-15 | 87.67% | 60.65% |
| Wu et al. [75] | WRN-28-10 | 88.25% | 60.04% |
| Zhang et al. [83] | WRN-28-10 | 89.36% | 59.64% |

## G  Societal impact

Neural networks are being deployed in a wide variety of applications ranging from ranking content on the web [15] to autonomous driving [5] via medical diagnostics [22]. As such, it is increasingly important to ensure that deployed models are robust and generalize to various input perturbations. While research on model robustness is welcome for safety critical applications, it is important to note that robustness can sometimes have unforeseen consequences. In particular, training robust models can lead to models that are overly insensitive to input variations [68] and that can increase bias [12]. It is also reported that adversarial robustness may not only be at odds with accuracy [69], but may also be at odds with privacy [63].

This work also introduces the use of generated data to improve adversarial robustness. The underlying generative models may leak confidential and private data [13] if they have been trained on a separate dataset. We protect against this by training generative models from scratch on the same data that is used to train our adversarially robust models.

Finally, our work is the first to match the performance of models trained with additional external data extracted from the "80 Million Tiny Images" dataset (80M-Ti) using only the original Cifar-10 dataset. Since the 80M-Ti contains some derogatory terms as categories and offensive images, it has been withdrawn. As such, we have made our generated datasets available online to allow researchers to avoid the use of 80M-Ti.

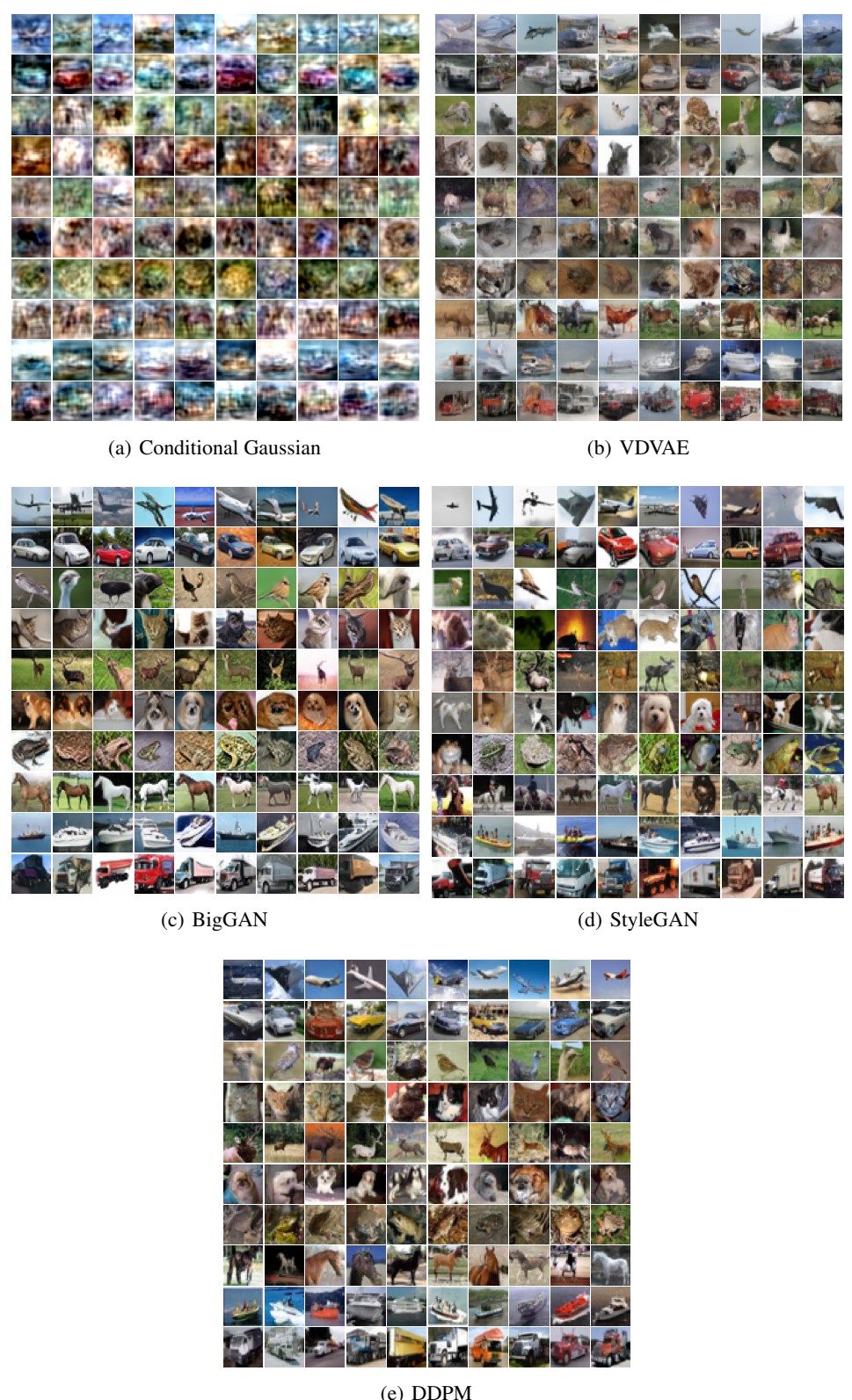

(a) Conditional Gaussian

(b) VDVAE

(c) BigGAN

(d) StyleGAN

(e) DDPM

Figure 12: CIFAR-10 samples generated by different approaches and used as additional data to train adversarially robust models. Each row correspond to a different class in the following order: airplane, automobile, bird, cat, deer, dog, frog, horse, ship, truck. Each image is assigned a *pseudo-label* using a standard classifier trained on the CIFAR-10 train set.