# OpenReview forum: "Improving Robustness using Generated Data"
_NeurIPS.cc/2021/Conference — NeurIPS 2021 Poster_

### Official Review · Reviewer_w69m · 2021-07-11

**Rating:** 7
**Confidence:** 4

**Summary:**

The paper proposes to improve robustness of neural networks against adversarial attacks without using any additional data. For it, they make use of samples generated by a generative model as their proxy for additional data. They want the additional samples to be diverse enough, and list additional conditions needed for robustness to improve using generated data. Empiricall they obtain strong results, often performing better than the methods that use additional data.


**Limitations And Societal Impact:**

The paper works on making networks more robust which will have a  positive societal impact.

**Main Review:**

Pros -
I like the idea of using generated samples instead of using additional real samples. The better performance of DDPM compared to other generative models does suggest that better generative models should lead to better improvements.  The authors do a good job in explaining the idea nicely along with good ablation studies. The strong results do suggest the efficacy of the idea.

Cons -

1. In order to truly realize the potential of the idea, the authors should try to test on other datasets for which it is difficult to easily get additional data. For example - some medical datasets or other datasets like DTD[1].

2. I don't agree with the 'Quality of Gnerative Models' paragraph in section 6.1. Using the test samples during training, should naturally improve the test set robustness. The authors should rather experiment with other datasets for which the generative process is known, such as CLEVER dataset [2] and show that as more samples are added from the true data distribution, the robustness increases.

3. It seems surprising that even with a low accuracy of the non-robust classifier, the robustness of the model increases. Can the authors justify/shed more light on this?

4. For Table 1 can you show results with stronger augmentations such as RandAugment.

[1] Cimpoi et al. Describing Textures in the Wild
[2] Johnson et al. Clevr: A diagnostic dataset for compositional language and elementary visual reasoning

[2]



**Time Spent Reviewing:**

3

---

> ### Author Response · Authors · 2021-08-09
> **Thank you for the review**
>
> Thank you for the review. We have addressed all the comments as follows:
>
> > In order to truly realize the potential of the idea, the authors should try to test on Another datasets for which it is difficult to easily get additional data. For example - some medical datasets or other datasets like DTD[1].
>
> We have selected standard benchmarks used for testing robustness to lp-norm bounded perturbations. These benchmarks allow us to directly compare with the state-of-the-art. We also note that CIFAR-10 only contains 50K images. We understand that adding more results on more datasets are always preferable. As such, we are now running experiments on the WILDS Camelyon dataset which is a medical imaging task consisting of classifying cancerous cell slides. However, since we need to retrain new DDPM models and re-run adversarial training experiments, we have not been able to obtain any results before the end of the rebuttal period.
>
> > I don't agree with the 'Quality of Gnerative Models' paragraph in section 6.1. Using the test samples during training, should naturally improve the test set robustness. The authors should rather experiment with other datasets for which the generative process is known, such as CLEVER dataset [2] and show that as more samples are added from the true data distribution, the robustness increases.
>
> While the experiments in Fig 4(b) and (c) are idealized situations, we still believe that they provide valuable insights:
>
> 1. Fig 4(b) shows that when Condition 2 is satisfied, the difference between models reduces and capacity takes a secondary role (since all models can bring their adversarial risk close to zero).
> 2. Fig 4(c) shows that when Condition 4 is satisfied (and Condition 2 is not), capacity matters greatly. We can observe that the larger models benefit more from increased coverage.
> 3. Both figures point to the fact that the quality of the generative model becomes less important when the capacity of the classifiers increases (as along as coverage is sufficient).
>
> We understand the concerns raised and have now re-run the same two experiments using our generative models to model the underlying "true" data distribution (i.e., models do not have access to the test dataset). In particular, during training, we use samples generated from a mixture of our class-conditional Gaussian and BigGAN distributions; during testing, we evaluate on a separate subset of 10K BigGAN samples. To evaluate Condition 2 (and replicate Fig 4(b)), we change the proportion of samples from the class-conditional Gaussian from 100% to 0% (thereby increasing the matching between train and test distributions). To evaluate Condition 4 (and replicate Fig 4(c)), we keep the ratio of samples from the class-conditional Gaussian distribution fixed at 99% and use the remaining 1% to include BigGAN samples corresponding to either 1, 2, … or 10 classes (thereby increasing coverage). We have updated the figures in the main manuscript and moved the old Fig 4(b) and (c) to the appendix. The new results are below (the robust accuracy is valuated against AA+MT at $\epsilon = 8/255$):
>
> **Matching**
>
> | Proportion of BigGAN samples | 0% | 1% | 2% | 5% | 10% | 20% | 50% | 100% |
> |---|---|---|---|---|---|---|---|---|
> | ResNet-18 | 31.29% | 62.46% | 73.42% | 83.39% | 87.39% | 89.98% | 92.27% | 93.26% |
> | WRN-70-16 | 32.55% | 79.83% | 85.53% | 89.91% | 91.99% | 93.44% | 94.91% | 95.50% |
>
> **Coverage**
>
> | Number of classes from BigGAN | 0 | 1 | 2 | 3 | 4 | 5 | 6 | 7 | 8 | 9 | 10 |
> |---|---|---|---|---|---|---|---|---|---|---|---|
> | ResNet-18 | 31.29% | 32.04% | 35.97% | 33.02% | 33.51% | 36.80% | 40.51% | 44.35% | 50.80% | 55.88% | 62.46% |
> | WRN-70-16 | 32.55% | 36.01% | 39.06% | 37.82% | 37.84% | 42.27% | 47.39% | 53.88% | 62.08% | 69.29% | 79.83% |
>
> **The conclusions remain unchanged.**
>
> > It seems surprising that even with a low accuracy of the non-robust classifier, the robustness of the model increases. Can the authors justify/shed more light on this?
>
> While this may seem surprising, it can be explained as follows. Without additional data (generated or real), the robust accuracy on CIFAR-10 obtained by a WRN-28-10 is 54.44%. By using a non-robust pseudo-labeling classifier that achieves an accuracy higher that 54.44%, we can construct a new training dataset that if classified correctly could lead to a robust accuracy higher than 54.44%.
>
> > For Table 1 can you show results with stronger augmentations such as RandAugment.
>
> We have now added additional augmentations (Cutout, CutMix, AutoAugment and RandAugment) in Table 1 and show identical trends to mixup. All augmentations produce samples that are too close to the train set and too far from the test set, which indicates that when when they provide samples that could complement the train set, these samples are far from the true distribution:
>
> | Setup | Complementary: Train | Test | Self | Coverage: Train | Test |
> |---|---|---|---|---|---|
> | mixup | 90.34% | 3.91% | 5.75% | 90.43% | 45.61% |
> | Cutout | 65.46% | 3.47% | 31.07% | 76.76% | 45.24% |
> | CutMix | 60.30% | 7.40% | 32.30% | 66.05% | 45.63% |
> | AutoAugment | 67.13% | 6.00% | 26.87% | 69.44% | 45.67% |
> | RandAugment | 61.23% | 8.85% | 29.92% | 65.51% | 45.78% |
>
> Apart from the aforementioned changes, we:
> * **Improved our results on CIFAR-10** against l-infinity perturbations of size 8/255 by using more generated samples and now reach 66.10% (improvement of +8.96% w.r.t. SOTA which also improves slightly on the SOTA robust accuracy obtained when using external data extracted from TinyImages-80M which stands at 65.87%). This improvement constitutes the largest absolute improvement ever made in this setting (since the work by Madry et al. in 2017).
> * Added results when using **StyleGAN** generated samples. These demonstrate that despite having significantly better FID and IS than the DDPM, the complementarity and coverage metrics computed in Table 1 are worse and result in lower performance for the StyleGAN.
> * Added results against **TinyImageNet-200** which demonstrate that the method can scale.
> See comments to other reviewers for more details.

---

> > ### Comment · Reviewer_w69m · 2021-08-15
> > **Follow up review**
> >
> > I thank the authors for addressing all my mentioned concerns. I will be willing to update my ratings once the experiment over the WILDS Camelyon dataset is done.

---

> > > ### Author Response · Authors · 2021-08-19
> > > **Thank you**
> > >
> > > Thank you for the time and effort spent reviewing our manuscript. The concerns raised on our initial submission have helped improve the paper. We are grateful.
> > >
> > > In the meantime, we have been able to train a new DDPM and run our method on the WILDS Camelyon17 dataset. Since we are interested in the in-distribution setting, we have focused our evaluation on the in-distribution validation set. This dataset provides a binary label (i.e., is there a tumor present?) and the training set contains 302,400 images (coming from 3 hospitals). It turns out that this dataset is not as challenging as we had hoped (i.e., as it is composed of a large amount of high-quality, standardized images), hence the improvements provided by our method, while still statistically significant, are small.
> > >
> > > Over 3 independent runs, without any additional generated samples, a WRN-28-10 achieves a robust accuracy of 85.55% $\pm$ 0.02% (against $\epsilon_\infty = 8/255$) and a standard accuracy of 92.81% $\pm$ 0.02%.
> > >
> > > With 1M generated samples (3x times the original dataset only), both robust and standard accuracy improve to 85.70% $\pm$ 0.01% and 93.03% $\pm$ 0.01%, respectively.

---

### Official Review · Reviewer_ZXq4 · 2021-07-13

**Rating:** 7
**Confidence:** 4

**Summary:**

This paper examines whether using additional generated data can improve the robustness through adversarial training.  They propose several conditions for generated data to best improve the performance of adversarial training and validate the impact of these conditions empirically.  They evaluate the performance of several generative models (DDPM, VDVAE, BigGAN, Class conditional gaussian) on improving robust performance and demonstrate that using samples from DDPM, they are able to outperform many models using external data in training.

**Limitations And Societal Impact:**

Limitations are addressed well through section 4.2.  For a larger dataset such as CelebA or ImageNet, are current state-of-the-art generative models able to generate data with good coverage/complementarity to improve adversarial training on these datasets?

**Main Review:**

Originality: The authors explore the possibility of using generated data to improve adversarial training which has not been explored before.  I think the contributions can be made more clear though since the generative models and adversarial training methods used are existing techniques.  For instance, I'm not very convinced that the second bullet in the contributions in the introduction is actually a contribution of this paper since DDPM has the lowest FID score.

Quality:  The depth of the experiments is very good.  The authors propose several conditions which may affect the performance of how much generated samples can improve robustness and provide experiments to demonstrate.  Some comments on the experiments:
- In Figure 4, I was wondering if the results could also be put in comparison with the performance of the models with corresponding architecture trained without additional generated data.  When the conditions are not satisfied well is there a point in which using the generated data can harm robust performance?
- In Section 6.1 and Figure 4b, I was wondering if the evaluation on the test set also includes the test set samples included in the training.  In that case, it is not very surprising that increasing the test-to-generated ratio since those examples were also used during the training.
- Is there a relationship between FID and Inception score with coverage and complementarity that are being measured?  I think these 2 metrics should be explained in the text.
- Could a column be added to Table 1 to compare the adversarial robustness of training with the same amount of additional data from each set of augmented and generated samples to validate the use of coverage and complementarity in assessing how beneficial the dataset is for adversarial training?

Clarity: The writing is clear.

Significance:  I think the main significance behind using generated data is that adversarial training has higher sample complexity and gathering this extra data for training can be costly.  Training a generator on a smaller dataset and then generating the extra data needed is more feasible.  The authors demonstrate that this generated data is able to improve robustness.

**Time Spent Reviewing:**

3

---

> ### Author Response · Authors · 2021-08-09
> **Thank you for the review**
>
> Thank you for the thoughtful review. We have addressed all the comments as follows:
>
> > Originality: The authors explore the possibility of using generated data to improve adversarial training which has not been explored before. I think the contributions can be made more clear though since the generative models and adversarial training methods used are existing techniques. For instance, I'm not very convinced that the second bullet in the contributions in the introduction is actually a contribution of this paper since DDPM has the lowest FID score.
>
> As stated, the effect of generated data on lp-norm bounded robustness has not been explored before. The fact that improvements upward of 6% are possible on such a competitive benchmark (i.e., CIFAR-10 against l-infinity perturbations of size 8/255) is worth sharing with the community. We also point out that, in the meantime, we were able to improve our results on CIFAR-10 against l-infinity perturbations of size 8/255 by using more generated samples and now reach 66.10% (an improvement of +8.96% w.r.t. SOTA which also improves slightly on the SOTA robust accuracy obtained when using external data extracted from TinyImages-80M which stands at 65.87%). **This improvement constitutes the largest improvement ever made in this setting** (since the work by Madry et al. in 2017).
>
> Furthermore, this work contributes to the community in the following ways: (1) we demonstrate that generated data can improve robustness even when the generated distribution is far from the real data distribution; (2) we provide a theoretical justification for it, and (3) we provide thorough experimental results (both in the main text and appendix).
>
> We also note that FID in itself does not measure complementarity to the training set: a generative model that perfectly memorizes the train set will obtain a very low FID (and high IS). We believe that the extra analysis from Table 1 is helpful and provides complementary information. For example, the BigGAN and VDVAE have very different FIDs but result in similar neighbor distributions that translate in similar robust performance (see Fig. 5). We have now also added results using a StyleGAN generative model which achieves better FID (2.57) but worse robust accuracy (58.17%) than the DDPM.
>
> > In Figure 4, I was wondering if the results could also be put in comparison with the performance of the models with corresponding architecture trained without additional generated data. When the conditions are not satisfied well is there a point in which using the generated data can harm robust performance?
>
> Given a properly tuned mixing ratio ($\alpha$), generated samples should not hurt performance (in the extreme, $\alpha$ can be set to 1). Furthermore, in the infinite-capacity regime, the performance (i.e., adversarial risk) on the training set will not be hurt either (if there are no conflicting labels). That being said, in the limited-capacity regime with an untuned $\alpha$, generated samples can definitely hurt performance as shown in Fig 5 (when small values of $\alpha$).
>
> Table 2 also shows the performance of some of these models without using generated samples: a WRN-70-16 reaches 57.14% robust accuracy. A WRN-28-10 reaches 54.44% robust accuracy without generated samples (Fig 5) and a ResNet-18 reaches 50.64% robust accuracy (not shown in the paper). We have added this information to Table 2 for completeness.
>
> > In Section 6.1 and Figure 4b, I was wondering if the evaluation on the test set also includes the test set samples included in the training. In that case, it is not very surprising that increasing the test-to-generated ratio since those examples were also used during the training.
>
> While the experiments in Fig 4(b) and (c) are idealized situations, we still believe that they provide valuable insights:
>
> 1. Fig 4(b) shows that when Condition 2 is satisfied, the difference between models reduces and capacity takes a secondary role (since all models can bring their adversarial risk close to zero).
> 2. Fig 4(c) shows that when Condition 4 is satisfied (and Condition 2 is not), capacity matters greatly. We can observe that the larger models benefit more from increased coverage.
> 3. Both figures point to the fact that the quality of the generative model becomes less important when the capacity of the classifiers increases (as along as coverage is sufficient).
>
> We understand the concerns raised and have now re-run the same two experiments using our generative models to model the underlying "true" data distribution (i.e., models do not have access to the test dataset). In particular, during training, we use samples generated from a mixture of our class-conditional Gaussian and BigGAN distributions; during testing, we evaluate on a separate subset of 10K BigGAN samples. To evaluate Condition 2 (and replicate Fig 4(b)), we change the proportion of samples from the class-conditional Gaussian from 100% to 0% (thereby increasing the matching between train and test distributions). To evaluate Condition 4 (and replicate Fig 4(c)), we keep the ratio of samples from the class-conditional Gaussian distribution fixed at 99% and use the remaining 1% to include BigGAN samples corresponding to either 1, 2, … or 10 classes (thereby increasing coverage). We have updated the figures in the main manuscript and moved the old Fig 4(b) and (c) to the appendix. The new results are below (the robust accuracy is valuated against AA+MT at $\epsilon = 8/255$):
>
> **Matching**
>
> | Proportion of BigGAN samples | 0% | 1% | 2% | 5% | 10% | 20% | 50% | 100% |
> |---|---|---|---|---|---|---|---|---|
> | ResNet-18 | 31.29% | 62.46% | 73.42% | 83.39% | 87.39% | 89.98% | 92.27% | 93.26% |
> | WRN-70-16 | 32.55% | 79.83% | 85.53% | 89.91% | 91.99% | 93.44% | 94.91% | 95.50% |
>
> **Coverage**
>
> | Number of classes from BigGAN | 0 | 1 | 2 | 3 | 4 | 5 | 6 | 7 | 8 | 9 | 10 |
> |---|---|---|---|---|---|---|---|---|---|---|---|
> | ResNet-18 | 31.29% | 32.04% | 35.97% | 33.02% | 33.51% | 36.80% | 40.51% | 44.35% | 50.80% | 55.88% | 62.46% |
> | WRN-70-16 | 32.55% | 36.01% | 39.06% | 37.82% | 37.84% | 42.27% | 47.39% | 53.88% | 62.08% | 69.29% | 79.83% |
>
> **The conclusions remain unchanged.**
>
> > Is there a relationship between FID and Inception score with coverage and complementarity that are being measured? I think these 2 metrics should be explained in the text.
>
> There is a loose relationship, however, the coverage and complementary metrics from Table 1 provide additional information that is not captured by FID or IS. In particular, a generative model that memorizes the train set will produce almost perfect FID and IS scores, but will produce a neighbor distribution of 100% matching the train set, 0% matching the test set and 0% matching itself. This is far from the ideal distribution of 1/3,1/3, 1/3. Similarly, a generative model that focuses on a subset of the true distribution can produce high IS, but low coverage (as exemplified by the BigGAN).  We added an explanation about these differences in the Appendix D (new last paragraph). Ultimately, it is important to look at a combination of various metrics. To illustrate our point, we have now **added results on StyleGAN** generated images which reach an FID of 2.57 and IS of 10.07 that are significantly better than the DDPM scores but have a slightly worse neighbor distribution (indicating a slight memorization of the training set) leading to worse robust performance:
>
> | Setup | Complementary: Train | Test | Self | Coverage: Train | Test | Robust Accuracy |
> |---|---|---|---|---|---|---|
> | StyleGAN | 28.13% | 27.22% | 44.65% | 50.16% | 48.29% | 58.17% |
> | DDPM | 29.29% | 29.17% | 41.54% | 49.07% | 49.10% | 60.73% |
>
> > Could a column be added to Table 1 to compare the adversarial robustness of training with the same amount of additional data from each set of augmented and generated samples to validate the use of coverage and complementarity in assessing how beneficial the dataset is for adversarial training?
>
> These results are visible in Fig 5, where we use 1M samples from each generative model. Augmentation techniques such as mixup, Cutout, RandAugment, AutoAugment have all been shown to not improve adversarial robustness as they all have low complementarity to the training set [28, 57, 72]. We've added an extra column with the requested information.
>
> Apart from the aforementioned changes, we:
> * Added results against **TinyImageNet-200** which demonstrate that the method can scale.
> See comments to other reviewers for more details.

---

> > ### Comment · Reviewer_ZXq4 · 2021-08-14
> > **Thank you for the response**
> >
> > Thank you for the response.  I believe that my main concerns were addressed and am raising my score to 7.
> >
> > In summary, my main concerns were:
> > 1. clarity of contributions - after the author's response to my review as well as reviewers r53y and kSDW, I believe the contributions are as follows
> > - first work to combine using generated samples with adversarial training and demonstrate that even using low quality generated samples/pseudolabels can improve robustness
> > - theoretical results about what conditions should be present to improve robustness as well as experiments showing what happens when these conditions are relaxed
> > - Complementary and coverage to assess quality of the samples from each generative model in improving robustness (I initially had doubts about this but this has been clarified, see 3)
> > - +8.96% increase in robust accuracy over SOTA models not using additional real data (TI-500k) and even improve over the SOTA model using the TI-500k dataset by +0.23%
> > 2. section 6.1 results are potentially a result of test set peeking - the authors have repeated these experiments using generative models and performed evaluations on a separate subset of generated samples than the ones used in training and have shown that the same trends hold
> > 3. relation between complementary and coverage metrics to FID and IS, it was unclear that FID and IS are insufficient metrics for measuring how helpful the generated images are for robustness with generative models tested - the authors have provided results for StyleGAN which has better FID and IS compared to DDPM but worse complementary and coverage and show that StyleGAN samples lead to lower robust accuracy.

---

> > > ### Author Response · Authors · 2021-08-19
> > > **Thank you**
> > >
> > > Thank you for the time and effort spent reviewing our manuscript. The concerns raised on our initial submission have helped improve the paper. We are grateful.

---

### Official Review · Reviewer_r53y · 2021-07-15

**Rating:** 6
**Confidence:** 4

**Summary:**

This work shows that data augmentation using good generative models helps improve adversarial robustness. Various experiments are conducted to show the importance of the quality of the generated images. The proposed method achieves state-of-the-art adversarial accuracy on CIFAR-10 and CIFAR-100 datasets, while the clean accuracy is not sacrificed.

**Limitations And Societal Impact:**

Limitations
- Listed above

Societal Impact
- N/A

**Main Review:**

Pros
- The proposed method is simple yet works well empirically. It achieves about 3~4% gain on adversarial accuracy compared to the previous state-of-the-art, even though the clean accuracy remains the same or also improves.
- The paper provides thorough experiments to back up their intuition that the quality of the generated images is important.

Cons
- The proposed method is a simple combination of two well-known works, adversarial training and DDPM.
- The scalability of the proposed method is concerning. The experiments on the paper (e.g., Figure 5) show that whether the generated images help or not depends heavily on their quality. In fact, DDPM is the only generative method that provides notable performance gain. Therefore, would the proposed method work on more large-scale datasets such as ImageNet where generating quality images is much harder?
- Some of the experiments are misleading. The paper provides Figure 4 (b) and (c) to prove that the quality and the coverage of the generative model is important. To show this, the paper considers an ideal generative model which is able to sample from the test dataset. However, 'looking at the test dataset directly' is very different from 'matching the true data distribution'.
- The appendix is not the final version. In fact, it includes comments which seem to include the authors' names. I worry this might be a violation of the double-blind review process.

=============== Update ===============

I thank the authors for their thoughtful response. Most of my concerns are addressed. However, I keep the concern that the method may not be scalable, even though the authors provide some additional experiments on slightly larger images (64$\times$64). I suggest the authors to add ImageNet results on future updates. That said, I appreciate that the proposed method achieves surprising empirical performance on the considered datasets, and increase my score to 6.

**Time Spent Reviewing:**

4

---

> ### Author Response · Authors · 2021-08-09
> **Thank you for the review**
>
> Thank you for the review. We have addressed all the comments as follows:
>
> > It achieves about 3~4% gain on adversarial accuracy compared to the previous state-of-the-art, even though the clean accuracy remains the same or also improves.
>
> This is a slight misstatement: on the very competitive CIFAR-10 l-infinity benchmark (at $\epsilon = 8/255$) our proposed method improves performance by 6.44%. This is the second largest improvement in this setting since the work by Madry et al. in 2017. In addition to this, the method generalizes to other similarly sized datasets with improvements between 3 and 5%. We point out that, in the meantime, we were able to improve our results on CIFAR-10 against l-infinity perturbations of size 8/255 by using more generated samples and now reach 66.10% (an improvement of +8.96% w.r.t. SOTA which also improves slightly on the SOTA robust accuracy obtained when using external data extracted from TinyImages-80M which stands at 65.87%). **This improvement constitutes the largest improvement ever made in this setting** (since the work by Madry et al. in 2017).
>
> > The proposed method is a simple combination of two well-known works, adversarial training and DDPM.
>
> Many published works are combinations of techniques and we take pride in the fact that the method is simple. To the best of our knowledge, this paper is the first to explore the use of generated data within robust training and provide a solid set of results that demonstrate large improvements in robust accuracy. Furthermore, we believe that this work contributes to the community in the following ways: (1) we demonstrate that generated data can improve robustness even when the generated distribution is far from the real data distribution; (2) we also provide a theoretical justification for it, and (3) we provide thorough experimental results (both in the main text and appendix).
>
> > The scalability of the proposed method is concerning. The experiments on the paper (e.g., Figure 5) show that whether the generated images help or not depends heavily on their quality. In fact, DDPM is the only generative method that provides notable performance gain. Therefore, would the proposed method work on more large-scale datasets such as ImageNet where generating quality images is much harder?
>
> I wouldn't qualify the issue of scalability as concerning. Recent work on diffusion and score-based models have, since we submitted our manuscript, been scaled to larger datasets (e.g., Dhariwal and Nichol, 2021) and their quality has also improved on CIFAR-10 (e.g., Vahdat et al, 2021). The main contribution of this paper is in demonstrating that generated data can improve robustness. We agree that the DDPM provides larger improvements compared to BigGAN or VD-VAE, but BigGAN and VD-VAE samples provide significant improvements nevertheless (well clear of the threshold for statistical significance). I also would like to remind the reviewer that robustness on CIFAR-10 against lp-norm bounded perturbations is still far from being solved and there is still value in providing improvement on CIFAR-10. To ease the raised concerns, we have now added results on **TinyImageNet** (64x64 images with 200 classes), where we improve upon classical adversarial training by +3.68%. We believe that given enough capacity, improvements on ImageNet are possible.
>
> > Some of the experiments are misleading. The paper provides Figure 4 (b) and (c) to prove that the quality and the coverage of the generative model is important. To show this, the paper considers an ideal generative model which is able to sample from the test dataset. However, 'looking at the test dataset directly' is very different from 'matching the true data distribution'.
>
> While the experiments in Fig 4(b) and (c) are idealized situations, we still believe that they provide valuable insights:
>
> 1. Fig 4(b) shows that when Condition 2 is satisfied, the difference between models reduces and capacity takes a secondary role (since all models can bring their adversarial risk close to zero).
> 2. Fig 4(c) shows that when Condition 4 is satisfied (and Condition 2 is not), capacity matters greatly. We can observe that the larger models benefit more from increased coverage.
> 3. Both figures point to the fact that the quality of the generative model becomes less important when the capacity of the classifiers increases (as along as coverage is sufficient).
>
> We understand the concerns raised and have now re-run the same two experiments using our generative models to model the underlying "true" data distribution (i.e., models do not have access to the test dataset). In particular, during training, we use samples generated from a mixture of our class-conditional Gaussian and BigGAN distributions; during testing, we evaluate on a separate subset of 10K BigGAN samples. To evaluate Condition 2 (and replicate Fig 4(b)), we change the proportion of samples from the class-conditional Gaussian from 100% to 0% (thereby increasing the matching between train and test distributions). To evaluate Condition 4 (and replicate Fig 4(c)), we keep the ratio of samples from the class-conditional Gaussian distribution fixed at 99% and use the remaining 1% to include BigGAN samples corresponding to either 1, 2, … or 10 classes (thereby increasing coverage). We have updated the figures in the main manuscript and moved the old Fig 4(b) and (c) to the appendix. The new results are below (the robust accuracy is valuated against AA+MT at $\epsilon = 8/255$):
>
> **Matching**
>
> | Proportion of BigGAN samples | 0% | 1% | 2% | 5% | 10% | 20% | 50% | 100% |
> |---|---|---|---|---|---|---|---|---|
> | ResNet-18 | 31.29% | 62.46% | 73.42% | 83.39% | 87.39% | 89.98% | 92.27% | 93.26% |
> | WRN-70-16 | 32.55% | 79.83% | 85.53% | 89.91% | 91.99% | 93.44% | 94.91% | 95.50% |
>
> **Coverage**
>
> | Number of classes from BigGAN | 0 | 1 | 2 | 3 | 4 | 5 | 6 | 7 | 8 | 9 | 10 |
> |---|---|---|---|---|---|---|---|---|---|---|---|
> | ResNet-18 | 31.29% | 32.04% | 35.97% | 33.02% | 33.51% | 36.80% | 40.51% | 44.35% | 50.80% | 55.88% | 62.46% |
> | WRN-70-16 | 32.55% | 36.01% | 39.06% | 37.82% | 37.84% | 42.27% | 47.39% | 53.88% | 62.08% | 69.29% | 79.83% |
>
> **The conclusions remain unchanged.**
>
> > The appendix is not the final version. In fact, it includes comments which seem to include the authors' names. I worry this might be a violation of the double-blind review process.
>
> We apologize for this mistake. The appendix should not have been part of the main PDF. The correct full version is in the supplementary material. We reached out to the PCs immediately upon noticing our mistake and were told that "reviewers are told to ignore any content beyond what was allowed in the main draft".
>
> Apart from the aforementioned changes, we:
> * Added results using **StyleGAN** generated samples. These demonstrate that despite having significantly better FID and IS than the DDPM, the complementarity and coverage metrics computed in Table 1 are worse and result in lower performance for the StyleGAN.
> See comments to other reviewers for more details.

---

> > ### Comment · Reviewer_r53y · 2021-08-19
> > **Thank you for the response.**
> >
> > I thank the authors for their thoughtful response. Most of my concerns are addressed. However, I keep the concern that the method may not be scalable, even though the authors provide some additional experiments on slightly larger images (64$\times$64). I suggest the authors to add ImageNet results on future updates. That said, I appreciate that the proposed method achieves surprising empirical performance on the considered datasets, and increase my score to 6.

---

> > > ### Author Response · Authors · 2021-08-19
> > > **Thank you**
> > >
> > > Thank you for the time and effort spent reviewing this paper. We would like to point out that our method is agnostic to the underlying generative model. While we agree that larger images and more complex datasets are likely to require better generative models (or classifiers with more capacity), the proposed method is general and should accommodate newer approaches when they become available.

---

### Official Review · Reviewer_kSDW · 2021-07-25

**Rating:** 6
**Confidence:** 4

**Summary:**

This paper shows experimentally that training on generated data (from a generative model not trained on external data) can improve adversarial robustness.  Surprisingly, even a class-conditional Gaussian generative model can improve adversarial robustness. The paper additionally provides sufficient conditions for generated data to help adversarial robustness.

**Limitations And Societal Impact:**

Yes

**Main Review:**

The main idea (generated data for adversarial robustness) is a cool and potentially powerful idea. Unfortunately, this idea's execution in this paper has many issues. In particular, the theory part has some problems, both technical as well as not much insight beyond "better pseudo-labeling models help" and "better generative models help".

Specific issues:
 - The claim "Complementarity and diversity should improve robust generalization" is not evaluated/tested in section 3.2; only that generated data improves robust generalization.
 - Although the authors claim to prove the results, I cannot find them in section 4. Some propositions would make section 4 more clear.
 - In the theory part of the paper, script and non-script letters (e.g. W, X, S) are interchanged in an incorrect way.
 - I believe the indicator function notation used throughout the paper is incorrect (see wikipedia). Consider using Iverson bracket notation.
 - Condition 4 is either ill-stated or never true. The empty set will have measure 0 under \hat{\mu} even though the empty set is measurable. Furthermore, the statement "There must be a non-zero probability of sampling any point" is not true for any distribution with a pdf (such as Gaussians).
 - "original-to-generated ratio" should be something like "proportion of data that is original"
 - I think the complementarity and coverage metrics (table 1) aren't properly defined. For example, what does "To estimate complementarity, we report the proportion of samples with a nearest neighbor in either the train set, test set or the sampled set itself." mean? What other sets could the nearest neighbor be in?
 - The appendix in the main pdf is unfinished (many comments from the authors scattered around) which potentially breaks anonymity.

In summary, the core idea and main result are great, but I think the paper needs refinement, especially for technical statements.












**Time Spent Reviewing:**

2.5

---

> ### Author Response · Authors · 2021-08-09
> **Thank you for the review**
>
> Thank you for the review. We believe that we have addressed all the technical issues appropriately.
>
> > not much insight beyond "better pseudo-labeling models help" and "better generative models help"
>
> We disagree. First, we do not think that the community would have predicted that improvements as large as 6.44% on the very competitive CIFAR-10 were possible using generated data.  Second, we expose the intricate relationship between generated data quality and classifier capacity in this setting. Third, the experiments show that **even bad pseudo-labeling models and bad generative models can still help robustness** (with improvements that are well clear of the threshold for statistical significance in all settings). Yes, "better pseudo-labeling models help" and "better generative models help", but these are not necessary components as both theoretical and experimental justifications suggest.
>
> > The claim "Complementarity and diversity should improve robust generalization" is not evaluated/tested in section 3.2; only that generated data improves robust generalization.
>
> We rephrased Section 3.2 to clarify that (1) data augmentation techniques (MixUp, Cutout, AutoAugment, RandAugment) which lack complementarity to the training set (as shown in Table 1) do not improve robustness (already shown by Rice et al. and Gowal et al.); (2) generating non-diverse data (e.g., generating images that comes from the same mode of the data distribution, for example always augmenting the same image from train or test set) intrinsically limits the ability of the model to generalize. We also changed the title of this section as suggested to "Generated data can improve robust generalization".
>
> > Although the authors claim to prove the results, I cannot find them in section 4. Some propositions would make section 4 more clear.
>
> To keep the text concise and readable, we avoided adding propositions and made use of footnotes. We've now re-factored the section to improve clarity. In particular, we now highlight the proofs (which are relegated to the appendix) more clearly. We note that they were already informally present in the text but were not highlighted sufficiently.
>
> **Proposition 1** shows that Conditions 1 and 2 (under the capacity-limited regime) are sufficient. Indeed, when $f_{\textrm{NR}}({\bf x}) = f^*({\bf x})$ for all ${\bf x} \in \mathcal{X}$ and $\hat{\mathcal{D}}$ is such that $\mu(\mathcal{W}) = \hat{\mu}(\mathcal{W})$ for all measurable $\mathcal{W} \subseteq \mathcal{X}$, Eq (4) and (5) become identical. Hence, their solutions achieve the same objective.
>
> **Proposition 2** shows that Conditions 1, 3, 4 (under the infinite-capacity regime) are sufficient. Indeed, Condition 3 guarantees that there are no images with conflicting labels within the perturbation set $\mathbb{S}$ of a realistic image (this extends the non-conflicting labels setup from Section 4.1). As such, it is possible to drive the objective from Eq (5) to zero. Since $f_{\textrm{NR}}({\bf x}) = f^*({\bf x})$ for all ${\bf x} \in \mathcal{X}$ (Condition 1) and the generated data covers the true distribution (Condition 4), the objective obtained from Eq (5) can only be zero if the objective to Eq (4) is also zero. Hence, the solutions of Eq (4) and Eq (5) achieve the same objective.
>
> We added a section in the appendix (new Appendix E) to highlight these more clearly. We also note that Condition 4 does not imply that the generative model is good and that violation to Condition 1 can still lead to big improvements in robust accuracy (see Fig 4(a)).
>
> > In the theory part of the paper, script and non-script letters (e.g. W, X, S) are interchanged in an incorrect way.
>
> They are now fixed. They should all appear in calligraphic font.
>
> > I believe the indicator function notation used throughout the paper is incorrect (see wikipedia). Consider using Iverson bracket notation.
>
> We apologize for the abuse of notation and are now using the Iverson bracket notation.
>
> > Condition 4 is either ill-stated or never true. The empty set will have measure 0 under \hat{\mu} even though the empty set is measurable. Furthermore, the statement "There must be a non-zero probability of sampling any point" is not true for any distribution with a pdf (such as Gaussians).
>
> We have now clarified that we mean measurable **non-empty** subsets (rather than just measurable). The statement "there must be a non-zero probability of sampling any point" was used to convey the meaning of the mathematical statement and we apologize for its imprecision. It has now been rephrased. Condition 4 now reads a follows:
>
> **Condition 4** (sufficient coverage): The likelihood of any finite sample in the set of realistic inputs $\mathcal{X}$ obtained from $\hat{\mathcal{D}}$ should be non-zero under the measure $\hat{\mu}$: $\hat{\mu}(\mathcal{W}) > 0$ for all non-empty measurable subsets $\mathcal{W} \subseteq \mathcal{X}$.
>
> > "original-to-generated ratio" should be something like "proportion of data that is original"
>
> We changed the x-axis label to the suggested version in all relevant plots.
>
> > I think the complementarity and coverage metrics (table 1) aren't properly defined. For example, what does "To estimate complementarity, we report the proportion of samples with a nearest neighbor in either the train set, test set or the sampled set itself." mean? What other sets could the nearest neighbor be in?
>
> Appendix D contains more details. We have now also added pseudo-code that explains precisely how the table is computed. In particular, we compute the ratio of neighbors that fall in the train, test or sampled set. For example, given 6 generated samples, the first sample's closest neighbor could be in the train set, the next two samples' closest neighbors could in the test set and the last three samples' closest neighbors could be in the set of generated samples (we do not match samples to themselves). This would result in ratios of 1/6, 1/3 and 1/2. If we had access the true data distribution, we would expect ratios of 1/3, 1/3, 1/3. We have added results on StyleGAN samples which demonstrate the utility of such an analysis (see list at the end of the response).
>
> > The appendix in the main pdf is unfinished (many comments from the authors scattered around) which potentially breaks anonymity.
>
> We apologize for this mistake. The appendix should not have been part of the main PDF. The correct full version is in the supplementary material. We reached out to the PCs immediately upon noticing our mistake and were told that "reviewers are told to ignore any content beyond what was allowed in the main draft".
>
> Apart from the aforementioned changes, we:
> * **Improved our results on CIFAR-10** against l-infinity perturbations of size 8/255 by using more generated samples and now reach 66.10% (improvement of +8.96% w.r.t. SOTA which also improves slightly on the SOTA robust accuracy obtained when using external data extracted from TinyImages-80M which stands at 65.87%). This improvement constitutes the largest absolute improvement ever made in this setting (since the work by Madry et al. in 2017).
> * Added results using **StyleGAN** generated samples. These demonstrate that despite having significantly better FID and IS than the DDPM, the complementarity and coverage metrics computed in Table 1 are worse and result in lower performance for the StyleGAN.
> * Added results against **TinyImageNet-200** which demonstrate that the method can scale.
> See comments to other reviewers for more details.

---

> > ### Comment · Reviewer_kSDW · 2021-08-15
> > **Thank you for your response**
> >
> > The authors have done a very good job with their response.
> >
> > I still have concerns with the definition of Condition 4, but I perhaps have a way to partially fix it. The issue is that for a distribution with a pdf, any set with a finite number of points will have 0 probability but will be non-empty. Additionally, any lower-dimensional manifold (lower compared to the dimension of the ambient space) will also have 0 probability but will be non-empty. In the case that the probability measure has a pdf, I think you might want to say "open measurable sets" instead of "non-empty measurable sets". I think this is equivalent to saying "for any point x in X, and for any non-zero radius r, if the ball with center at x and radius r is contained within X, then the ball has non-zero measure".
> >
> > I still don't find the analytic results very enlightening because the assumptions seem too strong, but the technical problems seem mostly fixed. The empirical contribution is quite interesting, so I'll raise my score to a 6.

---

> > > ### Author Response · Authors · 2021-08-19
> > > **Thank you**
> > >
> > > Thank you. We understood the raised concern and have modified Condition 4 accordingly. We appreciate the time and effort spent in helping us improve the paper.

---

### Decision · Program_Chairs · 2021-09-27

**Decision:**

Accept (Poster)

**Comment:**

All reviewers agree that the paper should be accepted. In addition, the empirical results are strong (substantial robustness gains on CIFAR-10).

It was discovered late in the review process that this paper makes use of the 80 million tiny images dataset, which has been retracted (https://groups.csail.mit.edu/vision/TinyImages/).  Following the NeurIPS ethical guidelines (https://neurips.cc/public/EthicsGuidelines), this dataset should not be used.  For this reason, the paper is being conditionally accepted.

UPDATE: The authors have revised the paper to satisfy the conditions and it has now been officially accepted.